# Warming and altered precipitation independently and interactively suppress alpine soil microbial growth in a decadal-long experiment

Yang Ruan[1,2], Ning Ling[1,2]*, Shengjing Jiang[1], Xin Jing[1], Jin-Sheng He[1,3]*, Qirong Shen[2], Zhibiao Nan[1]

[1]State Key Laboratory of Herbage Improvement and Grassland Agro-Ecosystems, College of Pastoral Agriculture Science and Technology, Lanzhou University, Lanzhou, China; [2]Jiangsu Provincial Key Lab for Solid Organic Waste Utilization, Jiangsu Collaborative Innovation Center for Solid Organic Waste Resource Utilization, Nanjing Agricultural University, Nanjing, China; [3]Institute of Ecology, College of Urban and Environmental Sciences, and Key Laboratory for Earth Surface Processes of the Ministry of Education, Peking University, Beijing, China

*For correspondence:
nling@njau.edu.cn (NL);
jshe@pku.edu.cn (J-SH)

Competing interest: The authors declare that no competing interests exist.

**Abstract** Warming and precipitation anomalies affect terrestrial carbon balance partly through altering microbial eco-physiological processes (e.g., growth and death) in soil. However, little is known about how such processes responds to simultaneous regime shifts in temperature and precipitation. We used the $^{18}$O-water quantitative stable isotope probing approach to estimate bacterial growth in alpine meadow soils of the Tibetan Plateau after a decade of warming and altered precipitation manipulation. Our results showed that the growth of major taxa was suppressed by the single and combined effects of temperature and precipitation, eliciting 40–90% of growth reduction of whole community. The antagonistic interactions of warming and altered precipitation on population growth were common (~70% taxa), represented by the weak antagonistic interactions of warming and drought, and the neutralizing effects of warming and wet. The members in *Solirubrobacter* and *Pseudonocardia* genera had high growth rates under changed climate regimes. These results are important to understand and predict the soil microbial dynamics in alpine meadow ecosystems suffering from multiple climate change factors.

## eLife assessment

This **important** study addresses the long-term effect of warming and precipitation on microbial growth, as a proxy for understanding the impact of global warming. The evidence that warming and altered precipitation exhibit antagonistic effects on bacterial growth is **compelling** and advances our understanding of microbial dynamics affected by environmental factors. This study will interest microbial ecologists, microbiologists, and scientists generally concerned with climate change.

## Introduction

Global climate change is threatening multi-dimensional ecosystem services, such as terrestrial primary productivity and soil carbon storage (*Jansson and Hofmockel, 2020*; *Walker et al., 2022*; *Zhou et al., 2022*), especially in high-elevation ecosystems (*Ma et al., 2017*; *Liu et al., 2018*). Of these, the effects of global climate change on microbial processes related to soil carbon cycling should

receive more extensive attention, because carbon balance will have feedbacks on climate system, and further reinforce/diminish the net impact on ecosystem functioning (*Jansson and Hofmockel, 2020*). Microbial growth and death, the critical eco-physiological processes, serve as the major engine of soil organic carbon (SOC) turnover and thus dominates the feedback on climate (*Sokol et al., 2022*). Quantitative estimates of trait-based responses of microbes to multiple climate factors is critical for improved biogeochemical models and predicting the feedback effects to global change.

Climate warming and precipitation regime shift can influence soil microbial physiological activities directly or indirectly (*Schimel, 2018*; *Jansson and Hofmockel, 2020*; *Purcell et al., 2022*; *Sokol et al., 2022*). The Tibetan Plateau is considered among the most sensitive ecosystems to climate change (*Liu et al., 2018*). In such alpine regions, warming can alleviate low temperature limitations to enzymatic activity, stimulating SOC mineralization and microbial respiration (*Dieleman et al., 2012*; *Streit et al., 2014*). Long-term warming reduces soil organic carbon pools and exacerbates carbon limitation of soil microbes, causing a negative effect on microbial growth and eco-physiological functions (*Jansson and Hofmockel, 2020*; *Melillo et al., 2017*; *Purcell et al., 2022*; *Streit et al., 2014*). Precipitation fluctuation constrains microbial physiological performance and functions, which is expected to be the major consequence of future climate change in mesic grassland ecosystems (*Cook et al., 2015*; *McHugh et al., 2017*; *Oppenheimer-Shaanan et al., 2022*; *Yuan et al., 2017*). Reduced precipitation affects soil processes notably by directly stressing soil organisms, and also altering the supply of substrates to microbes via dissolution, diffusion, and transport (*Schimel, 2018*). Increased frequency and magnitude of precipitation events could cause microbial species loss by 'filtering out' the taxa with low tolerance to increased soil moisture and drying-rewetting (*Evans and Wallenstein, 2014*). In addition, higher mean annual precipitation (MAP) triggers an increase in SOC decomposition (*Zhou et al., 2022*), which could cause a negative effect on microbial growth in long term. Collectively, climate change typically causes negative consequences on the microbe-associated processes in terrestrial ecosystems.

As temperature and precipitation are of particular relevance, the interactive effects of warming and altered precipitation remain largely illusive, especially on the population growth of soil microbes (*Zhu et al., 2016*; *Song et al., 2019*). Drought limits the resistance of the entire system to warming (*Hoeppner and Dukes, 2012*). Higher evapotranspiration in a warmer world will result in chronically lower average soil moisture (*Reich et al., 2018*), further reducing the eco-physiological performance of soil microbes (*Schimel, 2018*). In contrast, enhanced precipitation relieves overall water limitations caused by warming and improved primary productivity and soil respiration (*Fay et al., 2008*). The responses of microbial population growth to multiple climate factors could be complex because (i) the changed climate conditions can directly affect the eco-physiological characteristics of soil microbes and (ii) indirectly affect microbial functioning by altering soil physicochemical properties (e.g. redox conditions and nutrient allocation) and aboveground plant composition (*Qi et al., 2022*; *Yang et al., 2021*). The response of decomposer growth rates to the interaction of climate factors may be strongly idiosyncratic, varying among taxa, thus making predictions at the ecosystem level difficult.

The goal of current study is to comprehensively estimate taxon-specific growth responses of soil bacteria following a decade of warming and altered precipitation manipulation on the alpine grassland of the Tibetan Plateau, by using the $^{18}$O-quantitative stable isotope probing ($^{18}$O-qSIP) (*Figure 1A*). We focused on the single and interactive effects of temperature (T) and precipitation (P) on the population-specific growth of soil bacteria. We classified the interaction types as additive, synergistic, weak antagonistic, strong antagonistic and neutralizing interactions between climate factors (*Figure 1B*) by using the effect sizes and Hedges' *d* (an estimate of the standardized mean difference; *Côté et al., 2016*; *Harpole et al., 2011*; *Ma et al., 2019*; *Yue et al., 2017*). We addressed the following hypotheses: (1) long-term warming and altered precipitation regimes (i.e. drought or wet) have negative effects on microbial growth in alpine meadow soils; (2) the interactive effects between warming and altered precipitation on microbial population growth rates are not simply additive.

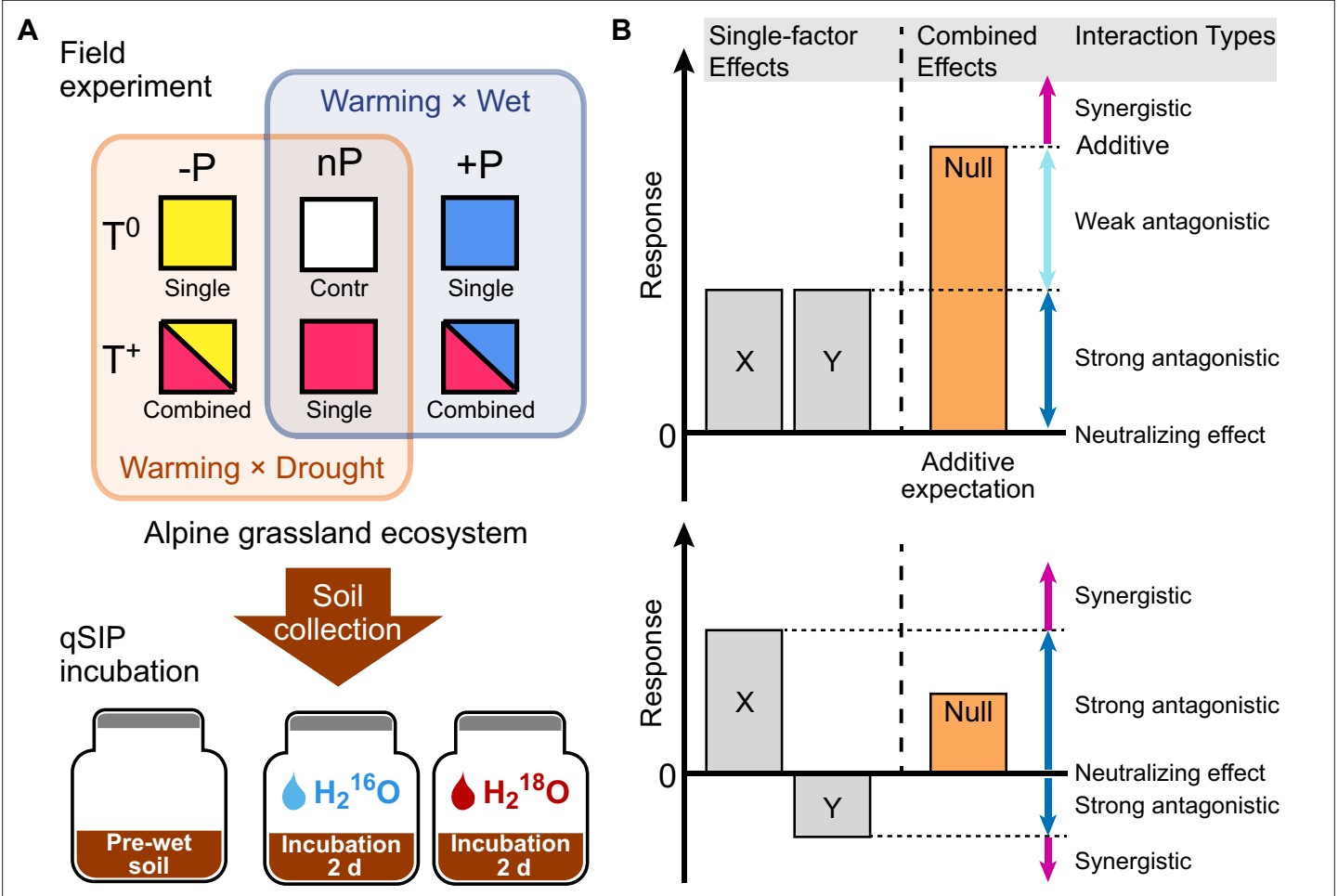

**Figure 1.** Field experiment design for simulated warming and altered precipitation, qSIP incubation, and the growth responses of soil bacteria to changing climate regimes. To examine the effects of warming and altered precipitation on an alpine grassland ecosystem, two levels of temperature ($T^0$, $T^+$), and three levels of precipitation (-P, nP, +P) were established in 2011. The soil samples were collected in 2020 and used for $^{18}O$-qSIP incubation (**A**). Potential interaction types between multiple climate factors (**B**). The diagram shows that two factors (X and Y) of warming and altered precipitation impact a biological response in the same direction (upper) or have opposing effects on when acting separately. Their combined effect could be additive, that is the sum of the two factor effects. Alternatively, the interaction types can be antagonistic or synergistic. Null model (we use the additive expectation as the null model here) provides the threshold for distinguishing between these interactions.

## Results

### Overall growth response of soil bacteria to warming and altered precipitation

Excess atom fraction $^{18}O$ value (***Figure 2***) and the population growth rate of each OTU were calculated using the qSIP pipeline. Collectively, 1373 OTUs were identified as ''$^{18}O$ incorporators'' (i.e. OTUs with growth rates significantly greater than zero) and used for subsequent data analyses. The maximum cumulative growth rates of the whole communities occurred in the ambient temperature and ambient precipitation condition ($T^0nP$), and all climate manipulations had negative effects on soil bacterial growth (***Figure 3A***). The individual impact of warming, drought, and wet conditions resulted in the most substantial negative effects on bacterial growth compared with the combined effects of warming × drought and warming × wet. A result that illustrates the antagonistic interactions between warming and modified precipitations patterns (***Figure 3B***). Moreover, the combined effect size of wet and warming was smaller than that of drought and warming, indicating a higher degree of antagonism of warming × wet.

Growth of the major bacterial phyla was also negatively influenced by the individual climate factors (***Figure 3C and D***). The antagonistic interactions of T and P were prevalent among the major phyla

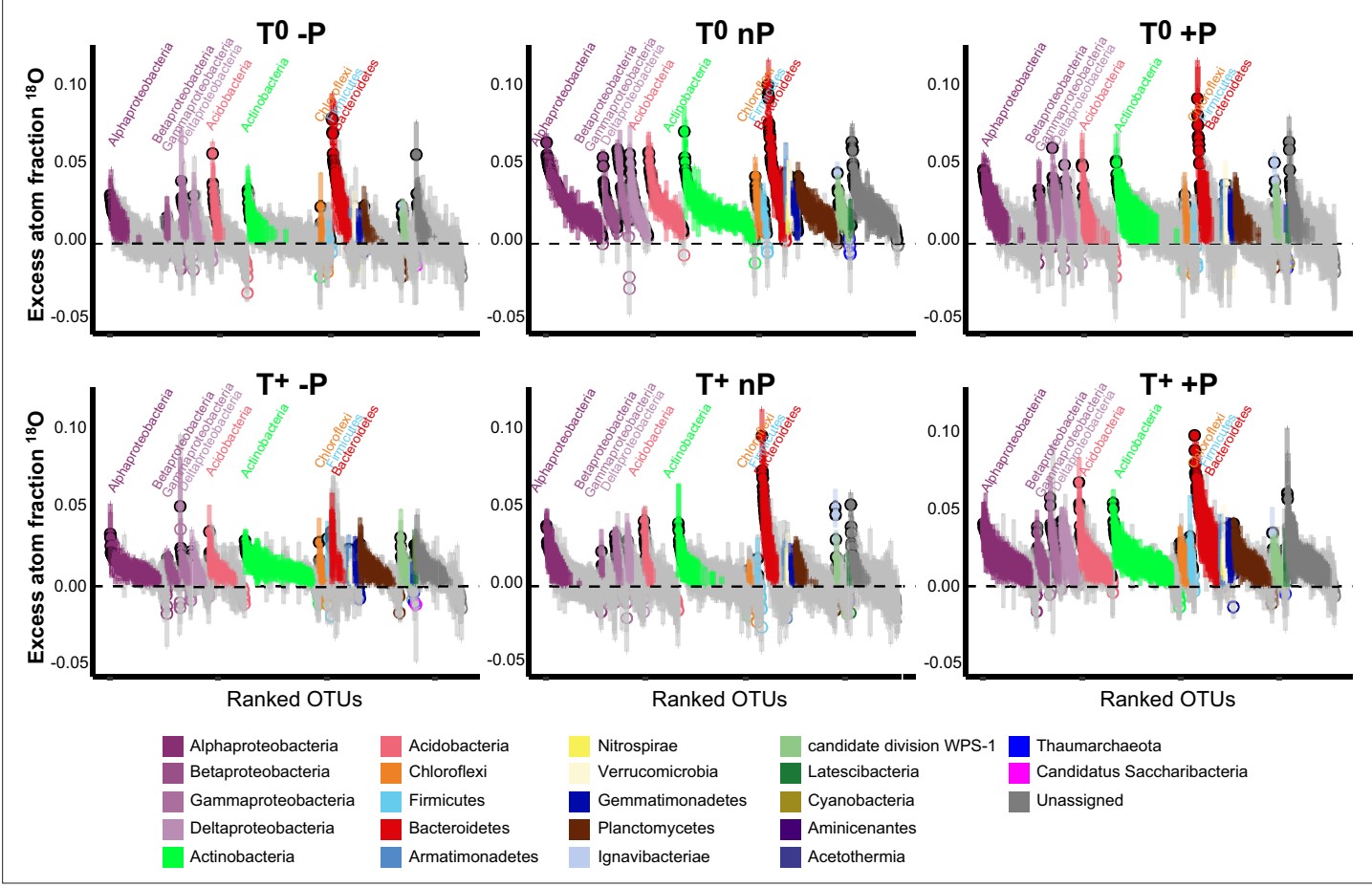

**Figure 2.** Species-specific shifts of $^{18}O$ excess atom fraction (EAF-$^{18}O$). Bars represent 95% confidence intervals (CIs) of OTUs. Each circle represents an OTU and color indicates phylum. The open circles with gray bars represent OTUs with 95% CI intersected with zero (indicating no significant $^{18}O$ enrichment); Closed circles represent the OTUs enriched $^{18}O$ significantly, whose 95% CIs were away from zero (i.e. the OTUs had detectable growth).

(except Bacteroidetes showed the additive interaction between drought and warming). We also found the significant smaller combined effect sizes of warming × wet in the major phyla compared with that of warming × drought (p < 0.05), such as Actinobacteria, Bacteroidetes and Betaproteobacteria, indicating higher degree of antagonism. In Actinobacteria and Bacteroidetes, the effect of wet and warming neutralized each other, as the combined effect of these two factors had no effect on growth.

## Phylogeny for the species whose growth subjected to different factor interactions

We constructed a phylogenetic tree including all $^{18}O$ incorporators in all six climate treatments (*Figure 4A*). The single-factor effects on the growth of incorporators tended to be negative (*Figure 4B*): Warming (T⁺nP) reduced the growth of 75% of the taxonomic groups, which was followed by drought and wet (74% and 67%, respectively). Warming × drought and warming × wet had the smaller impacts on the growth of incorporators, compared with the single effects (especially T⁺+P, only 43% of incorporators showed negative growth responses). The interaction type of T and P on the growth of ~70% incorporators was antagonistic (i.e. the combined effect size is smaller than the additive expectation) (*Figure 4C*). The weak antagonistic interaction on bacterial growth was dominant under the warming × drought conditions (41% of incorporators), while more incorporators (34%) whose growth subjected to neutralizing effect was found under the warming × wet conditions. These findings were robust at a subOTUs level by the zero-radius OTU (ZOTU) analysis (*Figure 3—figure supplement 1* and *Figure 4—figure supplement 1*).

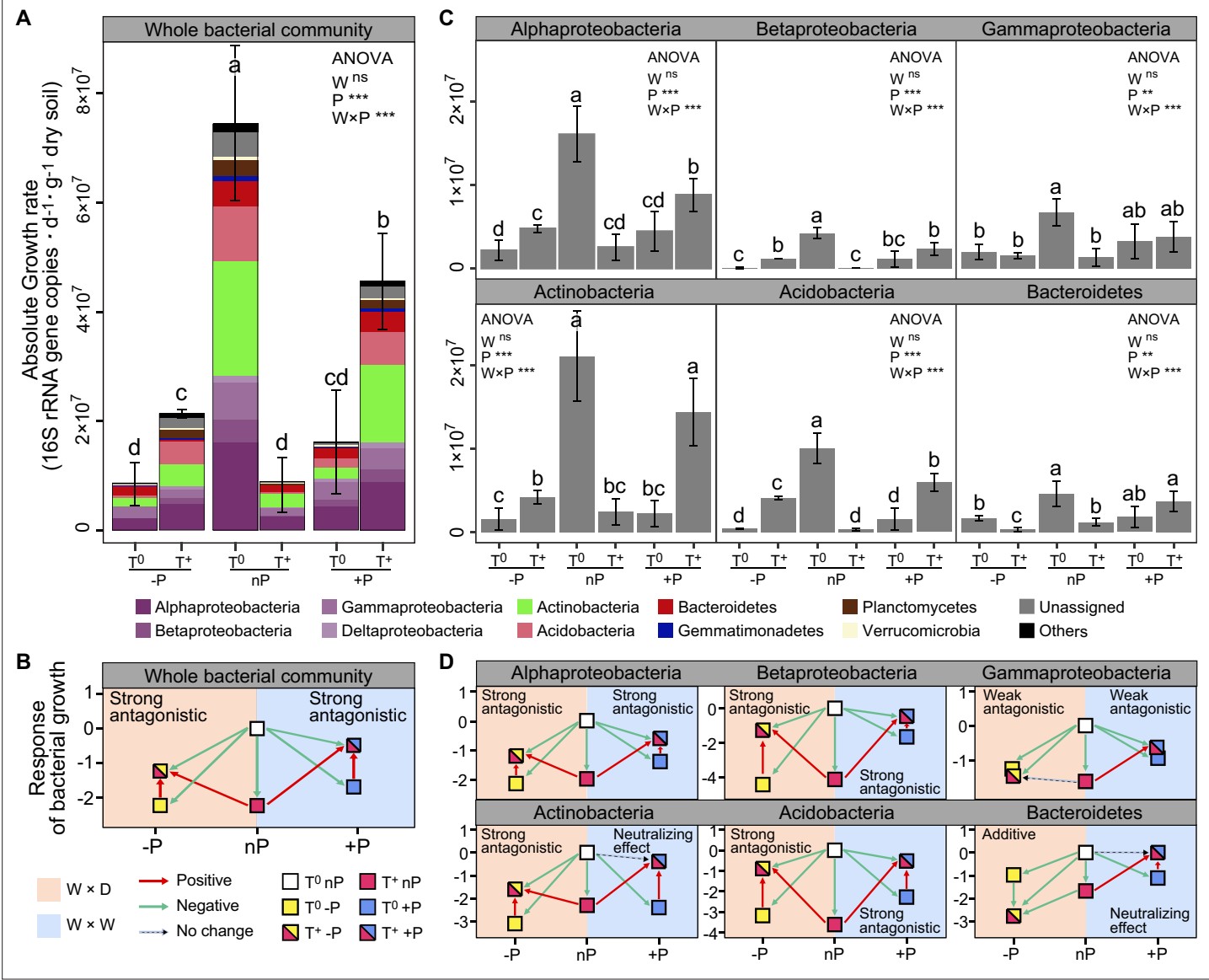

**Figure 3.** Bacterial growth responses to climate change and the interaction types between warming and altered precipitation. The growth rates (**A**), and responses of soil bacteria to warming and altered precipitation (**B**) at the whole community level. The growth rates (**C**), and responses of the dominant bacterial phyla (**D**) had similar trends with that of the whole community. Error bars depict means ± SD (n = 3). Different letters indicate significant differences between climate treatments (p < 0.05). The p-values were calculated using a two-tailed Student's *t*-test. Two-way ANOVA was used to examine the effects of climate factors on bacterial growth (**: p ≤ 0.01, ***: p ≤ 0.001, ns: no significance). 'W×P': the interaction effects of warming and altered precipitation; 'W×D': warming and drought scenario; 'W×W': warming and wet scenario.

The online version of this article includes the following figure supplement(s) for figure 3:

**Figure supplement 1.** The growth responses of grassland bacteria to warming and altered precipitation based on ZOTU (zero-radius OTU) analysis.

Phylogenetic relatedness can provide information on the ecological and evolutionary processes that influenced the emergence of the eco-physiological responses in taxonomic groups (*Evans and Wallenstein, 2014*). Nearest taxon index (NTI) was used to determine whether the species in a particular growth response are more phylogenetically related to one another than to other species (i.e. close or clustering on phylogenetic tree; *Figure 4—source data 1*). NTI is an indicator of the extent of terminal clustering, or clustering near the tips of the tree (*Evans and Wallenstein, 2014*; *Webb et al., 2002*). Overall, the most incorporators whose growth was influenced by the antagonistic interaction of T and P showed significant phylogenetic clustering (i.e. species clustered at the phylogenetic branches, indicating close genetic relationship; NTI > 0, p < 0.05). The incorporators whose growth

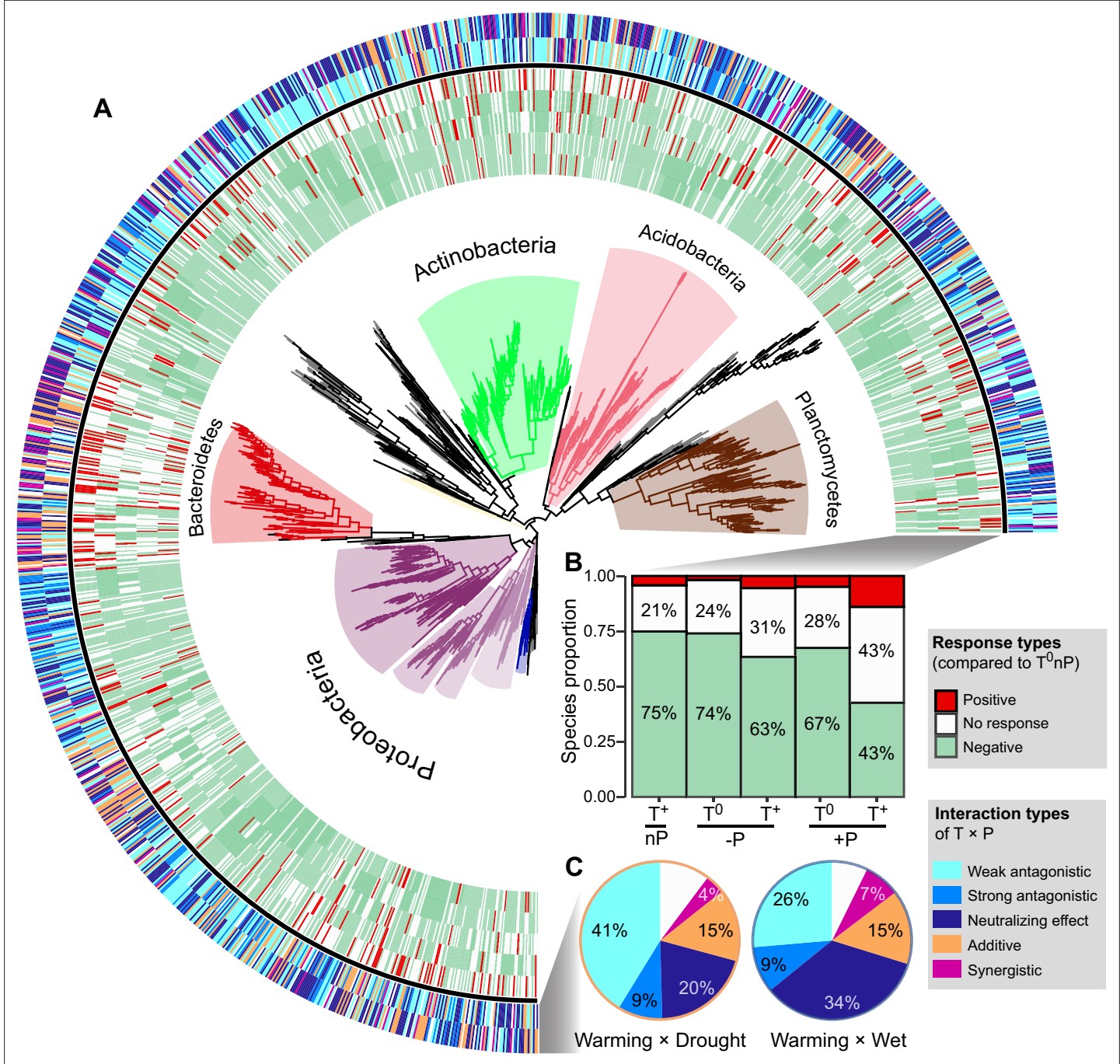

**Figure 4.** The growth responses and phylogenetic relationship of incorporators subjected to different interaction types under two climate scenarios. A phylogenetic tree of all incorporators observed in the grassland soils (**A**). The inner heatmap represents the single and combined factor effects of climate factors on species growth, by comparing with the growth rates in $T^0nP$. The outer heatmap represents the interaction types between warming and altered precipitation under two climate change scenarios. The proportions of positive or negative responses in species growth to single and combined manipulation of climate factors by summarizing the data from the inner heatmap (**B**). The proportions of species growth influenced by different interaction types under two climate change scenarios by summarizing the data from the outer heatmap (**C**).

The online version of this article includes the following source data and figure supplement(s) for figure 4:

**Source data 1.** The nearest taxon index (NTI) for incorporators subjected to different interaction types under two climate change scenarios.

**Figure supplement 1.** The growth responses of grassland bacteria at the genus level to warming and altered precipitation based on OTU analysis (**A** and **C**) and ZOTU analysis (**B** and **D**).

**Figure supplement 2.** The higher level of antagonism of wet × warming than that of drought × warming.

subjected to the additive interaction of warming × drought also showed significant phylogenetic clustering (p < 0.05), but randomly distributed under warming × wet scenario (p = 0.116). In addition, incorporators whose growth is influenced by the synergistic interaction of T and P showed random phylogenetical distribution under both climate scenarios (p > 0.05).

## Higher degree of antagonism in warming and wet scenario

We further assigned the antagonistic intensity to the five interaction types on a 5-point scale, from –1 to 3 for synergistic, additive, weak antagonistic, strong antagonistic and neutralizing effect, respectively (*Figure 4—figure supplement 2*), where the larger values represent higher degree of antagonism. Then, the overall antagonistic intensities of all incorporators under warming × drought and warming × wet scenarios were estimated by weighting the relative proportions of incorporators subjected to different interaction types (*Figure 4—figure supplement 2*). We found higher overall antagonistic intensity of warming × wet than that of warming × drought, contributing by a higher proportion of incorporators whose growth subjected to neutralizing effect (*Figure 4C* and *Figure 4—figure supplement 2*).

Of the total 1373 incorporators, 1218 were shared in both warming × drought and warming × wet scenarios (*Figure 5A*). That is, the difference in interactive effects between warming × drought and warming × wet we observed was due to a within-species change in growth response (i.e. phenotypic plasticity of organisms), rather than changes in species composition (i.e. species sorting). Of these species identified in both warming × drought and warming × wet scenarios, 453 incorporators were assigned a higher degree of antagonistic interaction of warming × wet than that of warming × drought. Further, the growth of 215 incorporators were influenced by the weak antagonistic interaction of warming × drought, and neutralizing effect of warming × wet. The growth response of these 215 species could contribute mainly to the overall growth patterns observed in grassland bacterial community under warming and altered precipitation scenarios, because of the prevalence of weak antagonistic interaction of warming × drought and neutralizing effect of warming × wet (*Figure 4C*).

We further assessed the potential functional traits of these 215 incorporators subjected to the dominant interaction types by PICRUST2 software (*Figure 5B*). The top three OTUs with the highest growth rates possessed extremely high species abundance (*Figure 5—source data 1*). The three taxa also possessed a higher functional potential related to carbon (C), nitrogen (N), sulfur (S), and phosphorus (P) cycling: the member affiliated to *Solirubrobacter* (OTU 14), has the high functional potential for aerobic C fixation and CO oxidation, nitrogen assimilation and assimilatory nitrite to ammonia, and phosphatase synthesis and phosphate transport transport-related functions. The members affiliated to the genus *Pseudonocardia* (OTU 5 and OTU 31), harbor a higher functional potential for aerobic C fixation, aerobic respiration, and CO oxidation, dissimilatory nitrate to nitrite and nitrogen assimilation, and sulfur mineralization functions. Furthermore, we annotated the genomic characteristics by aligning species sequences to the GTDB database (Genome Taxonomy Database), and we found that OTU 14 (*Solirubrobacter*) was predicted to have larger genomes and proteomes (*Figure 5—source data 1*). All these results suggested that these three species could play essential roles at the species and functional levels of ecosystems.

## Discussion

Microbial populations might respond differently to environmental changes, resulting in differential contributions to ensuing biogeochemical fluxes (*Blazewicz et al., 2020*). Here, we estimated microbial growth responses by using the qSIP technique to decadal-long warming and altered precipitation regimes in the alpine grassland ecosystem on the Tibetan Plateau, which is considered highly susceptible and vulnerable to climate change (*Ma et al., 2017*). After a decade of temperature and precipitation regime shift, the pervasive negative impacts of climate factors on soil bacterial growth in alpine grassland ecosystem were observed (*Figure 3*), which supports our first hypothesis that long-term warming and altered rainfall events consistently reduce microbial growth. Consistent with our findings, a recent experimental study demonstrated that 15 years of warming reduced the growth rate of soil bacteria in a montane meadow in northern Arizona (*Purcell et al., 2022*). These negative effects of climate factors on microbial growth are likely due to the variation related to availability of soil moisture and organic carbon (*Dieleman et al., 2012*; *Wu et al., 2011*). Previous evidences suggest

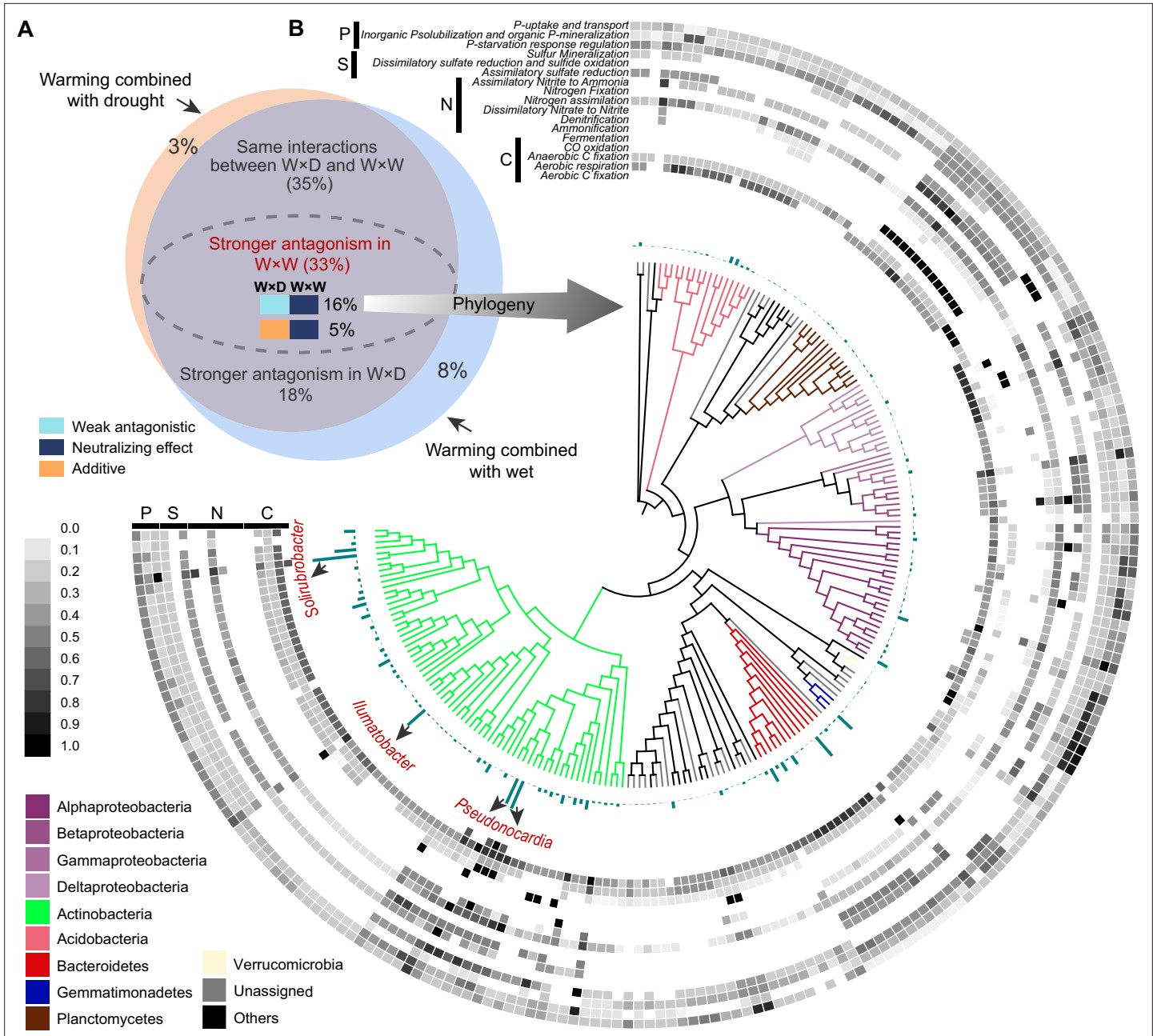

**Figure 5.** Within-species shift in interaction types contributed to the variance of the whole community growth response under two climate scenarios. Venn plots showing the overlaps of incorporators, and their interaction types between two climate scenarios (**A**). The phylogenetic relationship of the 215 incorporators whose growth dynamics were influenced by the weak antagonistic interaction of warming × drought and by the neutralizing effect of warming × wet (**B**). The blue-green bars represent the average growth rates of incorporators across different climate treatments. The heatmap displayed the potential functions associated with carbon and nutrient cycles predicted by Picrust2. The values of function potential were standardized (range: 0–1). 'W×D' represents warming × drought and 'W×W' represents warming × wet.

The online version of this article includes the following source data for figure 5:

**Source data 1.** Species and genomic information of the dominant active taxa in grassland soil under climate change conditions.

that warming has a negative impact on soil carbon pools (*Jansson and Hofmockel, 2020*; *Purcell et al., 2022*), mainly because of the rapid soil carbon mineralization and respiration (*Melillo et al., 2017*). Carbon is the critical element in cell synthesis, the reduction of microbially accessible carbon pools may explain the diminished microbial growth after long-term warming. In addition, long-term

warming can induce soil water deficiency (*Dieleman et al., 2012*; *Jansson and Hofmockel, 2020*), thereby slowing microbial growth.

Altered rainfall patterns, resulting in increased aridity or wetter conditions, mediate ecosystem cycling by affecting above- and below-ground biological processes (*Song et al., 2019*). As soil water availability is reduced, changes in osmotic pressure cause microbial death or dormancy, while others can accumulate solutes to survive under lower water potentials (*Schimel, 2018*). However, such accumulation of osmolytes could depend on highly energetic expenses (*Boot et al., 2013*; *Jansson and Hofmockel, 2020*; *Schimel et al., 2007*), resulting in less energetic allocation to growth (trade-offs between microbial growth and physiological maintenance). On the other hand, intensified rainfall patterns alter the composition and life strategies of soil bacteria, enriching the taxa with higher tolerance to frequent drying-rewetting cycles (*Evans and Wallenstein, 2014*). Such taxa may possess physiological acclimatization, such as synthesizing chaperones to stabilize proteins and thicker cell wall to withstand osmotic pressure (*Schimel et al., 2007*). These adaptation and acclimation strategies also create physiological costs (*Schimel et al., 2007*), increasing carbon allocation to physiological maintenance instead of new biomass (*Lipson, 2015*).

Climate-induced changes in the growth and structure of plant communities can also influence soil microbial growth by altering the amount and quality of plant-derived carbon (*Bardgett et al., 2013*). Increasing drought reduced the transfer of recently fixed plant carbon to soil bacteria and shifts the bacterial community towards slow growth and drought adaptation (*Fuchslueger et al., 2014*). A 17-year study of California grassland provided evidence that terrestrial net primary production (NPP) to precipitation gradient are hump-shaped, peaking when precipitation is near the multi-year mean growing season level (*Zhu et al., 2016*). Reduced NPP under increasing rainfall conditions could affect plant carbon inputs to the soil, ultimately having a negative effect on microbial growth.

Characterizing the interactive effects of multiple global change drivers on microbial physiological traits is important for predicting ecosystem responses and soil biogeochemical processes (*Song et al., 2019*; *Zhu et al., 2016*). In this study, a decade-long experiment revealed that bacterial growth in alpine meadows is primarily influenced by the antagonistic interaction between T and P (*Figures 3 and 4*). Similarly, a range of ecosystem processes have been revealed to be potentially subject to antagonistic interactions between climate factors, for instance, net primary productivity (*Shaw et al., 2002*), soil C storage and nutrient cycling processes (*Dieleman et al., 2012*; *Wu et al., 2011*; *Larsen et al., 2011*). Reduced precipitation can mute organic carbon mineralization by inhibiting soil respiration, which could maintain a relatively adequate soil carbon content and explain the diminished negative effects on microbial growth by the combined manipulation of warming and drought (*Jansson and Hofmockel, 2020*; *Wu et al., 2011*). Conversely, enhanced precipitation could stimulate SOM decomposition, causing further loss of soil carbon under warming conditions (*Zhou et al., 2022*). However, increased rainfall can also alleviate the moisture limitation on plant growth induced by warming, increasing plant-derived carbon inputs (*Jansson and Hofmockel, 2020*; *Wu et al., 2011*). The increased carbon inputs may alleviate microbial carbon limitation in soil, which partly explains the higher microbial growth rates under the combined treatment of warming and enhanced precipitation than that in the single climate factor treatments.

The degree of phylogenetic relatedness can indicate the processes that influenced community assembly, like the extent a community is shaped by environmental filtering (clustered by phylogeny) or competitive interactions (life strategy is phylogenetically random distribution) (*Evans and Wallenstein, 2014*; *Webb et al., 2002*). The results showed that the incorporators whose growth was influenced by the antagonistic interaction of T and P showed significant phylogenetic relatedness, indicating the occurrence of taxa more likely shaped by environment filtering (i.e. selection pressure caused by changes in temperature and moisture conditions). In contrast, the growing taxa affected by synergistic interactions of T and P showed random phylogenetic distributions (*Figure 4—source data 1*), which may be explained by competition between taxa with similar eco-physiological traits or changes in genotypes (possibly through horizontal gene transfer) (*Evans and Wallenstein, 2014*). We also found that the extent of phylogenetic relatedness to which taxa groups of T and P interaction types varied by climate scenarios, suggesting that different climate history processes influenced the ways bacteria survive temperature and moisture stress.

About one-third of bacterial species had growth with higher levels of antagonistic interaction of warming × wet than that of warming × drought (*Figure 5A*). By annotating the genomic information,

we further found that the species with the high growth rate (OTU 14, *Solirubrobacter*) has a relatively larger genome size and protein coding density (*Figure 5—source data 1*), indicating larger gene and function repertoires. A previous study showed that the genus *Solirubrobacter* detected in the Thar desert of India is involved in multiple biochemical processes, such as N and S cycling (*Sivakala et al., 2018*). Members in the genus *Solirubrobacter* are also considered to contribute positively to plant growth (*Liu et al., 2020*), and can be used to predict the degradation level of grasslands, indicating the critical roles on maintaining ecosystem services (*Yan et al., 2022*). This is, however, still to be verified, as the functional output from PICRUSt2 is less likely to resolve rare environment-specific functions (*Dieleman et al., 2012*). This suggests the development of methods combining qSIP with metagenomes and metatranscriptomes to assess the functional shifts of active microorganisms under global change scenarios. Note that the experimental parameters such as DNA extraction and PCR amplification efficiencies also have significant effects on the accuracy of growth assessment. This alerts the need to standardize experimental practices to ensure more realistic and reliable results.

The evaluation of ecosystem models based on results obtained from single-factor experiments usually overestimate or underestimate the impact of global change on ecosystems, because the interactions between climate factors may not be simply additive (*Dieleman et al., 2012*; *Wu et al., 2011*; *Zhou et al., 2022*). Our results demonstrated that both warming and altered precipitation negatively affect the growth of grassland bacteria; However, the combined effects of warming and altered precipitation on the growth of ~70% soil bacterial taxa were smaller than the single-factor effects, suggesting antagonistic interaction. This suggests the development of multifactor manipulation experiments in precise prediction of future ecosystem services and feedbacks under climate change scenarios.

## Materials and methods
### Study design and soil sampling
The warming-by-precipitation experiment was established in 2011 at the Haibei National Field Research Station of Alpine Grassland Ecosystem (37°37′N, 101°33′E, with elevation 3215 m), which is located on the northeastern Tibetan Plateau in Qinghai Province, China. The climate type is a continental monsoon with mean annual precipitation of 485 mm and the annual average temperature approximately –1.7°C. The high rainfall and temperature mainly occur in the peak-growing season (from July to August *Liu et al., 2018*). The soils are Mat-Gryic Cambisols, with the average pH value of surface soil (0–10 cm) being 6.4 (*Ma et al., 2017*).

The experimental design has been described previously in *Ma et al., 2017*. Briefly, experimental plots were established in an area of 50 m × 110 m in 2011, using a randomized block design with warming and altered precipitation treatments. Each block contained six plots (each plot was 1.8 m × 2.2 m), crossing two levels of temperature [ambient temperature ($T^0$), elevated temperature of top 5 cm layer of the soil by 2°C ($T^+$)], and three levels of precipitation [natural precipitation (nP, represents ambient condition), 50% reduced precipitation (-P, represents 'drought' condition) and 50% enhanced precipitation (+P, represents 'wet' condition)]. In the warming treatments, two infrared heaters (1000 mm length, 22 mm width) were suspended in parallel at 150 cm above the ground within each plot. Rain shelters were used to control the received precipitation in the experimental plots. Four 'V'-shaped transparent polycarbonate resin channels (Teijin Chemical, Japan) were fixed at a 15° angle, above the infrared heaters, to intercept 50% of incoming precipitation throughout the year. The collected rainfall from the drought plots was supplied to the wet plots manually after each precipitation event by sprinklers, increasing precipitation by 50%. To control for the effects of shading caused by infrared heaters, two 'dummy' infrared heaters and four 'dummy' transparent polycarbonate resin channels were installed in the control plots. Each treatment had six replicates, resulting in thirty-six plots.

Soil samples for qSIP incubation were collected in August 2020. Considering the cost of qSIP experiment (including the use of isotopes and the sequencing of a large number of DNA samples), we randomly selected three out of the six plots, serving as three replicates for each treatment. In each plot, three soil cores of the topsoil (0–5 cm in depth) were randomly collected and combined as a composite sample, which can be considered as a mixture of rhizosphere and bulk soils. Each sampling point was as far away from infrared heaters as possible to minimize the impact of physical shading on

the plants. The fresh soil samples were shipped to the laboratory and sieved (2 mm) to remove root fragments and stones.

## $^{18}$O-qSIP incubation

The incubations were similar to those reported in a previous study (*Ruan et al., 2023*). Soil samples of ambient temperature treatments (including T$^0$-P, T$^0$nP, and T$^0$ +P) were air-dried at 14°C (average temperature across the growth season), while the soil samples of warming treatments (including T$^+$-P, T$^+$nP, and T$^+$+P) were air-dried at 16°C (increased temperature of 2°C). There is no violent air convection in the incubator and the incubation temperature is relatively low (compared to 25°C used in previous studies), resulting slower evaporation and no significant discoloration caused by severe soil dehydration after 48 hr. A portion of the air-dried soil samples was taken as the pre-wet treatment (i.e., before incubation without H$_2$O addition). We incubated the air-dried soils (2.00 g) with 400 μl of 98 atom% H$_2$$^{18}$O ($^{18}$O treatment) or natural abundance water ($^{16}$O treatment) in the dark for 2 d by using sterile glass aerobic culture bottles (Diameter: 29 mm; Height: 54 mm). After incubation, soils were destructively sampled and stored at –80°C immediately. A total of 54 soil samples, including 18 pre-wet samples (6 treatments × 3 replicates) and 36 incubation samples (6 treatments × 3 replicates × 2 types of H$_2$O addition), were collected.

## DNA extraction and isopycnic centrifugation

Total DNA from all the collected soil samples was extracted using the FastDNA SPIN Kit for Soil (MP Biomedicals, Cleveland, OH, USA) according to the manufacturer's instructions. Briefly, the mechanical cell destruction was attained by multi-size beads beating at 6 m s$^{-1}$ for 40 s, and then FastDNA SPIN Kit for Soil (MP Biomedicals, Cleveland, OH, USA) was used for DNA extraction. All DNA samples were extracted by the same person within 2–3 hr, and a unifying procedure of cell lysis and DNA extraction was used. The concentration of extracted DNA was determined fluorometrically using Qubit DNA HS (High Sensitivity) Assay Kits (Thermo Scientific, Waltham, MA, USA) on a Qubit 4 fluorometer (Thermo Scientific, Waltham, MA, USA). The DNA samples of 2-d incubation were used for isopycnic centrifugation, according to a previous publication (*Ruan et al., 2023*). Briefly, 3 μg DNA were added into 1.85 g ml$^{-1}$ CsCl gradient buffer (0.1 M Tris-HCl, 0.1 M KCl, 1 mM EDTA, pH = 8.0) with a final buoyant density of 1.718 g ml$^{-1}$. Approximately 5.1 ml of the solution was transferred to an ultracentrifuge tube (Beckman Coulter QuickSeal, 13 mm × 51 mm) and heat-sealed. All tubes were spun in an Optima XPN-100 ultracentrifuge (Beckman Coulter) using a VTi 65.2 rotor at 177000 g at 18°C for 72 h with minimum acceleration and braking.

Immediately after centrifugation, the contents of each ultracentrifuge tube were separated into 20 fractions (~250 μl each fraction) by displacing the gradient medium with sterile water at the top of the tube using a syringe pump (Longer Pump, LSP01-2 A, China). The buoyant density of each fraction was measured using a digital hand-held refractometer (Reichert, Inc, Buffalo, NY, USA) from 10 μl volumes. Fractionated DNA was precipitated from CsCl by adding 500 μl 30% polyethylene glycol (PEG) 6000 and 1.6 M NaCl solution, incubated at 37°C for 1 hr and then washed twice with 70% ethanol. The DNA of each fraction was then dissolved in 30 μl of Tris-EDTA buffer.

## Quantitative PCR and sequencing

Total 16S rRNA gene copies for DNA samples of all the fractions were quantified using the primers for V4-V5 regions: 515F (5'-GTG CCA GCM GCC GCG G-3') and 907R (5'-CCG TCA ATT CMT TTR AGT TT-3') (*Guo et al., 2018*). The V4-V5 primer pairs were chosen to facilitate integration and comparison with data from previous studies (*Ruan et al., 2023*; *Zhang et al., 2016*). Plasmid standards were prepared by inserting a copy of purified PCR product from soil DNA into *Escherichia coli*. The *E. coli* was then cultured, followed by plasmid extraction and purification. The concentration of plasmid was measured using Qubit DNA HS Assay Kits. Standard curves were generated using 10-fold serial dilutions of the plasmid. Each reaction was performed in a 25 μl volume containing 12.5 μl SYBR Premix Ex Taq (TaKaRa Biotechnology, Otsu, Shiga, Japan), 0.5 μl of forward and reverse primers (10 μM), 0.5 μl of ROX Reference Dye II (50 ×), 1 μl of template DNA and 10 μl of sterile water. A two-step thermocycling procedure was performed, which consisted of 30 s at 95°C, followed by 40 cycles of 5 s at 95°C, 34 s at 60°C (at which time the fluorescence signal was collected). Following qPCR cycling, melting curves were conducted from 55 to 95°C with an increase of 0.5°C every 5 s to ensure that results were

representative of the target gene. Average PCR efficiency was 97% and the average slope was –3.38, with all standard curves having $R^2 \geq 0.99$.

The DNA of pre-wet soil samples (unfractionated) and the fractionated DNA of the fractions with buoyant density between 1.703 and 1.727 g ml$^{-1}$ (7 fractions) were selected for 16S rRNA gene sequencing by using the same primers of qPCR (515F/907R). The fractions with density between 1.703 and 1.727 g ml$^{-1}$ were selected because they contained more than 99% gene copy numbers of each sample. A total of 270 DNA samples [18 total DNA samples of prewet soil +252 fractionated DNA samples (6 treatments × 3 replicates × 2 types of water addition × 7 fractions)] were sequenced using the NovaSeq6000 platform (Genesky Biotechnologies, Shanghai, China).

The raw sequences were quality-filtered using the USEARCH v.11.0 (**Edgar, 2010**). In brief, the paired-end sequences were merged and quality filtered with 'fastq_mergepairs' and 'fastq_filter' commands, respectively. Sequences < 370 bp and total expected errors > 0.5 were removed. Next, 'fastx_uniques' command was implemented to identify the unique sequences. Subsequently, high-quality sequences were clustered into operational taxonomic units (OTUs) with 'cluster_otus' commandat a 97% identity threshold, and the most abundant sequence from each OTU was selected as a representative sequence. The taxonomic affiliation of the representative sequence was determined using the RDP classifier (version 16) (**Wang et al., 2007**). In total, 19,184,889 reads of the bacterial 16S rRNA gene and 6,938 OTUs were obtained. The sequences were uploaded to the National Genomics Data Center (NGDC) Genome Sequence Archive (GSA) with accession numbers CRA007230.

## Quantitative stable isotope probing calculations

As $^{18}$O labeling occurs during cell growth via DNA replication, the amount of $^{18}$O incorporated into DNA was used to estimate the growth rates of active taxa. The density shifts of OTUs between $^{16}$O and $^{18}$O treatments were calculated following the qSIP procedures (**Hungate et al., 2015**; **Koch et al., 2018**). Briefly, the number of 16S rRNA gene copies per taxon (e.g. genus or OTU) in each density fraction was calculated by multiplying the relative abundance (acquisition by sequencing) by the total number of 16S rRNA gene copies (acquisition by qPCR). Then, the GC content and molecular weight of a particular taxon were calculated. Further, the change in $^{18}$O isotopic composition of 16S rRNA genes for each taxon was estimated. We assumed an exponential growth model over the course of the incubations. The growth rate is a function of the rate of appearance of $^{18}$O-labeled 16S rRNA genes. Therefore, the growth rate of taxon $i$ was calculated as:

$$g_i = \ln\left(\frac{N_{\text{TOTAL}it}}{N_{\text{LIGHT}it}}\right) \times \frac{1}{t} \tag{1}$$

where $N_{\text{TOTAL}it}$ is the number of total gene copies for taxon $i$ and $N_{\text{LIGHT}it}$ represents the unlabeled 16S rRNA gene abundances of taxon $i$ at the end of the incubation period (time $t$). $N_{\text{LIGHT}it}$ is calculated by a function with four variables: $N_{\text{TOTAL}it}$, average molecular weights of DNA (taxon $i$) in the $^{16}$O treatment ($M_{\text{LIGHT}i}$) and in the $^{18}$O treatment ($M_{\text{LAB}i}$), and the maximum molecular weight of DNA that could result from assimilation of H$_2$$^{18}$O ($M_{\text{HEAVY}i}$) (**Koch et al., 2018**). We further calculated the average growth rates (represented by the production of new16S rRNA gene copies of each taxon per g dry soil per day) along the incubation, using the following equation (**Stone et al., 2021**):

$$\frac{dN_i}{dt} = N_{\text{TOTAL}it}(1 - e^{-g_it}) \times \frac{1}{t} \tag{2}$$

where $t$ is the incubation time (day). All data calculations were performed using the qSIP pipeline **Source code 1** in R (Version 3.6.2) (**Streit et al., 2014**).

## Single and combined effects of climate change factors

To address the effects of warming and altered precipitation on microbial growth rates, three single-factor effects (warming, 50% reduced precipitation only, and 50% enhanced precipitation only) and two combined effects (combined warming and reduced precipitation manipulation and combined warming and enhanced precipitation manipulation) were calculated by the natural logarithm of response ratio (lnRR), representing the response of microbial growth rates in the climate change

treatment compared with that in the ambient treatment (*Yue et al., 2017*). The lnRR for growth rates was calculated as:

$$\mathrm{lnRR} = \ln\left(\frac{X_t}{X_c}\right) \tag{3}$$

where $X_t$ is the observed growth rates in climate treatment and $X_c$ is that in control. 95% confidence interval (CI) was estimated using a bootstrapping procedure with 1000 iterations (*Ruan et al., 2023*). If the 95% CI did not overlap with zero, the effect of treatment on microbial growth is significant.

## The interaction between warming and altered precipitation

All six climate treatments were divided into two groups, warming combined with reduced precipitation scenario (Warming × Drought), and warming combined with enhanced precipitation scenario (Warming × Wet), by using the ambient temperature and precipitation treatment ($T^0nP$) as control (*Figure 1A*). Hedges' $d$, an estimate of the standardized mean difference, was used to assess the interaction effects of warming × drought (i.e. reduced precipitation) and warming × wet (i.e. enhanced precipitation), respectively (*Yue et al., 2017*). The interaction effect size ($d_I$) of warming × drought or warming × wet was calculated as:

$$d_I = \frac{(X_{AB} - X_A) - (X_B - X_c)}{2s} J(m) \tag{4}$$

where $X_c$, $X_A$, $X_B$, and $X_{AB}$ are growth rates in the control, treatment groups of factor A, B, and their combination (AB), respectively. 95% CI was estimated using a bootstrapping procedure with 1000 iterations. The $s$ and $J(m)$ are the pooled standard deviation and correction term for small sample bias, respectively, which were calculated by the following equations:

$$s = \sqrt{\frac{(n_c - 1)\,s_c^2 + (n_A - 1)\,s_A^2 + (n_B - 1)\,s_B^2 + (n_{AB} - 1)\,s_{AB}^2}{n_c + n_A + n_B + n_{AB} - 4}} \tag{5}$$

$$J(m) = 1 - \frac{3}{4\,(n_c + n_A + n_B + n_{AB} - 4) - 1} \tag{6}$$

where $n_c$, $n_A$, $n_B$, and $n_{AB}$ are the sample sizes, and $s_c$, $s_A$, $s_B$, and $s_{AB}$ are the standard deviations in the control, experimental groups of A, B, and their combination (AB), respectively.

The interaction types between warming and altered precipitation were mainly classified into three types, that is additive, synergistic and antagonistic, according to the single-factor effects and 95% CI of $d_I$. If the 95% CI of $d_I$ overlapped with zero, the interactive effect of warming and altered precipitation was additive. The synergistic interaction included two cases: (1) the upper limit of 95% CI of $d_I$ < 0 and the single-factor effects were either both negative or have opposite directions; (2) the lower limit of 95% CI of $d_I$ > 0 and both single-factor effects were positive. The antagonistic interaction also included two cases: (1) the upper limit of 95% CI of $d_I$ < 0 and both single-factor effects were positive; (2) the lower limit of 95% CI of $d_I$ > 0 and the single-factor effects were either both negative or have opposite directions (*Yue et al., 2017*). We further divided antagonistic interaction into three sub-categories: weak antagonistic interaction, strong antagonistic interaction, and neutralizing effect, by comparing the single-factor and combined effect sizes (*Figure 1B*). The weak antagonistic interaction determined if the combined effect size was larger than the single-factor effect sizes, but smaller than their expected additive effect. The strong antagonistic interaction determined if the combined effect size was smaller than the single-factor effect sizes but not equal to zero. The neutralizing effect represented the combined effect size is equal to zero, and at least one single-factor effect size is not equal to zero.

## Statistical analyses

Uncertainty of growth rates (95% CI) was estimated using a bootstrapping procedure with 1000 iterations (*Ruan et al., 2023*). The cumulative growth rates at the phylum-level were estimated as the sum of taxon-specific growth rates of those OTUs affiliated to the same phylum. Significant differences of bacterial growth rates for each group between climate treatments were assessed by two-way ANOVA in R (version 3.6.2). Phylogenetic trees were constructed in Galaxy /DengLab (http://mem.rcees.ac.cn:

8080) with PyNAST Alignment and FastTree functions (*Caporaso et al., 2010*; *Price et al., 2009*). The trees were visualized and edited using iTOL (*Letunic and Bork, 2016*). To estimate the phylogenetic patterns of incorporators whose growth subjected to different factor interaction types, the nearest taxon index (NTI) was calculated by the 'picante' package in R (version 3.6.2; *Webb et al., 2002*). NTI with values larger than 0 and their p values less than 0.05 represent phylogenetic clustering. The p values of NTI between 0.05 and 0.95 represent random phylogenetic distributions. KO gene annotation of taxa was performed by PICRUSt2 (Phylogenetic Investigation of Communities by Reconstruction of Unobserved States), which predicted functional abundances based on marker gene sequences (*Dieleman et al., 2012*). The marker genes related to carbon (C), nitrogen (N), sulfur (S), and phosphorus (P) cycling were selected according to the conclusions reported in previous documents (*Dai et al., 2020*; *Llorens-Marès et al., 2015*; *Nelson et al., 2015*).

## Acknowledgements

This work was supported by the National Science Foundation of China [42277100 (NL)].

## Additional information

### Funding

| Funder | Grant reference number | Author |
|---|---|---|
| National Science Foundation of China | 42277100 | Ning Ling |

The funders had no role in study design, data collection and interpretation, or the decision to submit the work for publication.

### Author contributions

Yang Ruan, Methodology, Writing - original draft, Writing – review and editing; Ning Ling, Conceptualization, Supervision, Writing – review and editing; Shengjing Jiang, Resources, Writing – review and editing; Xin Jing, Writing – review and editing; Jin-Sheng He, Qirong Shen, Zhibiao Nan, Resources, Project administration

### Author ORCIDs

Ning Ling ⓘ http://orcid.org/0000-0003-1250-4073
Jin-Sheng He ⓘ http://orcid.org/0000-0001-5081-3569

Reviewer #1 (Public review): https://doi.org/10.7554/eLife.89392.3.sa1
Author response https://doi.org/10.7554/eLife.89392.3.sa2

## Additional files

### Supplementary files
• MDAR checklist

• Source code 1. The qSIP calculation procedures and R code for calculating species growth rates.

### Data availability

The sequence data were uploaded to the National Genomics Data Center (NGDC) Genome Sequence Archive (GSA) with accession number CRA007230.

The following dataset was generated:

| Author(s) | Year | Dataset title | Dataset URL | Database and Identifier |
|---|---|---|---|---|
| Ruan Y, Ling N, Jiang SJ, Jing X, Shen QR, Nan ZB, He JS | 2023 | Soil microbiome under warming and altered precipitation | https://ngdc.cncb.ac.cn/gsa/browse/CRA007230 | National Genomics Data Center (NGDC), CRA007230 |

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
