## [Editor Report · eLife assessment]

This **important** study addresses the long-term effect of warming and precipitation on microbial growth, as a proxy for understanding the impact of global warming. The evidence that warming and altered precipitation exhibit antagonistic effects on bacterial growth is **compelling** and advances our understanding of microbial dynamics affected by environmental factors. This study will interest microbial ecologists, microbiologists, and scientists generally concerned with climate change.

---

## [Referee Report · Reviewer #1 (Public review)]

Warming and precipitation regime change significantly influences both above-ground and below-ground processes across Earth's ecosystems. Soil microbial communities, which underpin the biogeochemical processes that often shape ecosystem function, are no exception to this, and although research shows they can adapt to this warming, population dynamics and ecophysiological responses to these disturbances are not currently known. The Qinghai-Tibet Plateau, the Third Pole of the Earth, is considered among the most sensitive ecosystems to climate change. The manuscript described an integrated, trait-based understanding of these dynamics with the qSIP data. The experimental design and methods appear to be of sufficient quality. The data and analyses are of great value to the larger microbial ecological community and may help advance our understanding of how microbial systems will respond to global change. There are very few studies in which the growth rates of bacterial populations from multifactorial manipulation experiments on the Qinghai-Tibet Plateau have been investigated via qSIP, and the large quantity of data that comprises the study described in this manuscript, will substantially advance our knowledge of bacterial responses to warming and precipitation manipulations.

---

## [Author Response]

The following is the authors’ response to the original reviews.

**eLife assessment**
This valuable study addresses the long-term effect of warming and altered precipitation on microbial growth, as a proxy for understanding the impact of global warming. While the methods are compelling and the evidence supporting the claims is solid, additional analysis of the data would strengthen the study, which should be of broad interest to microbial ecologists and microbiologists.

We sincerely appreciate your assessment and thoughtful comments, which are valuable and very helpful for improving our manuscript. We have carefully considered all comments, and made extensive, thorough corrections and additional analysis of the data, which we hope to meet with approval.

**Reviewer #1 (Public Review):**
Warming and precipitation regime change significantly influences both above-ground and below-ground processes across Earth's ecosystems. Soil microbial communities, which underpin the biogeochemical processes that often shape ecosystem function, are no exception to this, and although research shows they can adapt to this warming, population dynamics and ecophysiological responses to these disturbances are not currently known. The Qinghai-Tibet Plateau, the Third Pole of the Earth, is considered among the most sensitive ecosystems to climate change. The manuscript described an integrated, trait-based understanding of these dynamics with the qSIP data. The experimental design and methods appear to be of sufficient quality. The data and analyses are of great value to the larger microbial ecological community and may help advance our understanding of how microbial systems will respond to global change. There are very few studies in which the growth rates of bacterial populations from multifactorial manipulation experiments on the Qinghai-Tibet Plateau have been investigated via qSIP, and the large quantity of data that comprises the study described in this manuscript, will substantially advance our knowledge of bacterial responses to warming and precipitation manipulations.

We appreciate the encouragement and positive comments.

Specific comments:(1) Please add some names of microbial groups with most common for the growth rates.

We have added the sentence “The members in Solirubrobacter and Pseudonocardia genera had high growth rates under changed climate regimes” In the Abstract (Line 57-58).

(2) L47-48, consider changing "microbial growth and death" to "microbial eco-physiological processes (e.g., growth and death)", and changing "such eco-physiological traits" to "such processes".

Done (Line 47 and 48).

(3) L50-51, the author estimated bacterial growth in alpine meadow soils of the Tibetan Plateau after warming and altered precipitation manipulation in situ. Actually, the soil samples were collected and incubated in the laboratory rather than in the field like the previous experiment conducted by Purcell et al. (2021, Global Change Biology). "In situ" would lead me to believe that the qSIP incubation was conducted in the field, so I think the use of the word in situ is inappropriate here. [https://onlinelibrary.wiley.com/doi/full/10.1111/gcb.15911]

Agreed. We have deleted “in situ”.

(4) L52, what does "interactive global change factors" mean?

We have revised this sentence to “the growth of major taxa was suppressed by the single and combined effects of temperature and precipitation” (Line 52-53).

(5) L61, in my opinion, "Microbial diversity" belongs to the category of species composition, rather than ecosystem functional services. Please revise it.

Agree. We have deleted it.

(6) L69, consider changing "further" to "thus".

Done (Line 70).

(7) L82, delete "The evidence is overwhelming that".

Done.

(8) L85-90, these two sentences have similar meanings, please express them concisely.

We have deleted the sentence “Altered precipitation, particularly drought or heavy precipitation events, also tends to negatively influence soil processes and biodiversity”.

(9) L91, the effect of drought on soil microorganisms is lacking here.

We have added the sentence “Reduced precipitation affects soil processes notably by directly stressing soil organisms, and also altering the supply of substrates to microbes via dissolution, diffusion, and transport” in the Introduction (Line87-89).

(10) L102, "Growth" should be highlighted here, as changes in relative abundance can also be classified as population dynamics. The use of the term "population dynamics" will eliminate the highlight of this study in calculating the growth rate of microbial species in in-situ soil based on qSIP. Consider changing "population dynamics" to "population-growth responses" or something like that.

Done (Line 98).

(11) L105, please note that this citation focuses on plant physiological characteristics.

We have revised the reference (Line 102).

(12) L115, "soil temperature, water availability" should be considered as a direct impact of climate change, rather than an indirect impact on microorganisms.

We have deleted them.

(13) L134-135, please clarify the interaction types between which climate factors.

We have deleted this sentence.

(14) L135-138, suggest modifying or deleting this sentence. The results in this study are already eco-physiological data and do not need to be further "understood and predicted".

We have deleted this sentence.

(15) L150, "The experimental design has been described in previously". I think this refers to another study and not the actual incubations in this study. Also in L198, suggest a change to "Incubation conditions were similar to those previously described". So, it's clear it's not the same experiment.

We have revised these sentences to “has been described previously in (Ma et al., 2017)” (Line 136) and “according to a previous publication” (Line 194).

Reference:

Ma, Z., Liu, H., Mi, Z., Zhang, Z., Wang, Y., Xu, W. et al. (2017). Climate warming reduces the temporal stability of plant community biomass production. Nature Communications, 8, 15378.

(16) L188, change "pre-wet soil samples" to "pre-wet samples" and change "soil samples for 48h incubation" to "incubation samples". What does "pre-wet" mean? Does it represent soil pre-cultivation?

Done. The pre-wet samples, i.e., the soil samples before incubation (T = 0 d), were used to estimate the initial microbial composition. "pre-wet" does not mean soil pre-cultivation. We have added the description “A portion of the air-dried soil samples was taken as the pre-wet treatment (i.e., before incubation without H2O addition)” in MATERIALS AND METHODS (Line 174-175).

(17) Unify the time unit of incubation (hour or day). Consider changing "48 h" to "2 d" in Materials and Methods.

Done.

(18) L247, what version of RDP Classifier was used?

We used RDP v16 database for taxonomic annotation. We have added this information in the revision (Line 246).

(19) L270, "average molecular weights".

Done (Line 268).

(20) L272-275, based on the preceding description, it appears that the culture period was limited to 48 hours. Please confirm it.

Apologize for this mistake. We have revised it (Line 273).

(21) L297, switch the order of the first two sentences of this paragraph.

Done (Line 297).

(22) L331, change "smaller-than-additive" to "smaller than their expected additive effect".

Done (Line 331).

(23) L374 and 381, I struggle with why "larger combined effects" than single factor effects represent higher degree of antoninism, and I think it should be "smaller combined effects".

Agree. We have revised it according to this suggestion (Line 369 and 374).

(24) L375, remove "than that of drought and warming".

Done.

(25) L405, simplify the expression, change "between different warming and rainfall regimes" to "between climate regimes"

We have deleted this sentence.

(26) L406-408, species are already on the phylogenetic tree and they can not "clustered at the phylogenetic branches", but the functional traits of microbes can. Please revise it.

We have revised this sentence to “Overall, the most incorporators whose growth was influenced by the antagonistic interaction of T and P showed significant phylogenetic clustering (i.e., species clustered at the phylogenetic branches; NTI > 0, p < 0.05)” (Line 402-404).

(27) L409, the same as above, and consider removing "The incorporators subjected to".We have revised this sentence to “The incorporators whose growth subjected to the additive interaction of warming × drought also showed significant phylogenetic clustering (p < 0.05)” (Line 404-406).(28) L412, consider changing "incorporators subjected to the synergistic interaction" to "the synergistic growth responses under multifactorial changes".

We have revised the sentence to “incorporators whose growth is influenced by the synergistic interaction showed phylogenetically random distribution under both climate scenarios (p > 0.05)” (Line 407-409).

(29) L505-506, please add a reference for this sentence.

Done (Line 488).

(30) L511-514, It should be noted that the production of MBC does not necessarily imply a net change in the C pool size. The accelerated growth rates may result in expedited turnover of MBC, rather than an increase in carbon sequestration.

Thanks. We have deleted this sentence.

(31) Language precision. In the discussion section there must be some additional caveats introduced to some of the claims the authors are making. For instance, L518, the author should clarify that "in this study, the bacterial growth in alpine grassland may be influenced by antagonistic interactions between multiple climatic factors after a decadal-long experiment". Because other studies may exhibit different results due to the focus on different ecosystem functions as well as environmental conditions. As such, softening of the language is recommended- lines are noted below- and these will not adjust the outcomes of this study, but support more precise interpretation.

We have revised the sentence to “In this study, a decade-long experiment revealed that bacterial growth in alpine meadows is primarily influenced by the antagonistic interaction between T and P” (Line 497-499).

(32) Picrust analysis is a good way to connect species and their functions, especially Picrust2, which updated the reference database and optimized the algorithm to improve its prediction accuracy (Douglas et al., 2020, Nature Biotechnology). However, the link between microbial taxonomy and microbial metabolism is still not straightforward, especially in diverse microbial communities like soils. The authors should introduce caveats within discussion that they know the limitations of their methods. For context, as a reader who does metabolisms in soils, I found myself somewhat disappointed when piecrust data was introduced and not properly caveated. Particularly, it might be helpful to introduce briefly in the last paragraph of the results. These caveats are necessary to not potentially overstate the author's findings, and to make sure the reader knows the authors understand the very clear limitations of these methods. [https://www.nature.com/articles/s41587-020-0548-6]

Thanks. We have introduced caveats in DISCUSSION, that is “This is, however, still to be verified, as the functional output from PICRUSt2 is less likely to resolve rare environment-specific functions (Douglas et al., 2020)” (Line 540-542).

Reference:

Douglas, G., Maffei, V., Zaneveld, J., Yurgel, S., Brown, J., Taylor, C. et al. (2020). PICRUSt2 for prediction of metagenome functions. Nature Biotechnology, 38, 1-5.

(33) Although the author has explained the potential causes for the negative effects of different climate change factors (i.e., warming, drought, and wet) on microbial growth, there seems to be a lack of a summary assertion and an extension on how climate change affects microbial growth and related ecosystem functions. It is recommended to make a general summary of the results in the last part of Discussion.

We have added a general summary in the last paragraph of DISCUSSION, that is “Our results demonstrated that both warming and altered precipitation negatively affect the growth of grassland bacteria; However, the combined effects of warming and altered precipitation on the growth of ~70% soil bacterial taxa were smaller than the single-factor effects, suggesting antagonistic interaction. This suggests the development of multifactor manipulation experiments in precise prediction of future ecosystem services and feedbacks under climate change scenarios” (Line 552-558).

(34) L546, please add the taxonomic information for "OTU 14".

Done (Line 533).

(35) L800, change "The phylogenetic tree" to "A phylogenetic tree".

Done (Line 762).

**Reviewer #2 (Public Review):**
Summary:The authors aimed to describe the effect of different temperature and precipitation regimes on microbial growth responses in an alpine grassland ecosystem using quantitative 18O stable isotope probing. It was found that all climate manipulations had negative effects on microbial growth, and that single-factor manipulations exerted larger negative effects as compared to combined-factor manipulations. The degree of antagonism between factors was analyzed in detail, as well as the differential effect of these divergent antagonistic responses on microbial taxa that incorporated the isotope. Finally, a hypothetical functional profiling was performed based on taxonomic affiliations. This work gives additional evidence that altered warming and precipitation regimes negatively impact microbial growth.Strengths:A long term experiment with a thorough experimental design in apparently field conditions is a plus for this work, making the results potentially generalisable to the alpine grassland ecosystem. Also, the implementation of a qSIP approach to determine microbial growth ensures that only active members of the community are assessed. Finally, particular attention was given to the interaction between factors and a robust approach was implemented to quantify the weight of the combined-factor manipulations on microbial growth.

We appreciate the reviewer’s positive comments.

Weaknesses:The methodology does not mention whether the samples taken for the incubations were rhizosphere soil, bulk soil or a mix between both type of soils. If the samples were taken from rhizosphere soil, I wonder how the plants were affected by the infrared heaters and if the resulting shadow (also in the controls with dummy heaters) had an effect on the plants and the root exudates of the parcels as compared to plants outside the blocks? If the samples were bulk soil, are the results generalisable for a grassland ecosystem? In my opinion, it is needed to add more info on the origin of the soil samples and how these were taken.

The samples taken for the incubations can be considered as a mixture of rhizosphere and bulk soils. During soil sampling, we did not use conventional rhizosphere soil collection methods. However, there is a certain proportion of fragmented roots in the soil samples we collected, indicating that soil properties are influenced by plants. We have added this description in MATERIALS AND METHODS (Line 158).

To minimize the impact of physical shading on the plants, each sampling point was as far away from infrared heaters as possible. We have added this information of soil collection in MATERIALS AND METHODS, that is “In each plot, three soil cores of the topsoil (0-5 cm in depth) were randomly collected and combined as a composite sample, which can be considered as a mixture of rhizosphere and bulk soils. Each sampling point was as far away from infrared heaters as possible to minimize the impact of physical shading on the plants. The fresh soil samples were shipped to the laboratory and sieved (2-mm) to remove root fragments and stones.” (Line 157-162).

Previous studies based on our field experiment assessed the effects of warming and altered precipitation on soil microbial communities (Zhang et al., 2016), the temporal stability of plant community biomass (Ma et al., 2017), shifting plant species composition and grassland primary production (Liu et al., 2018). These studies provide guidance for the experiment design and execution.

Reference:

Zhang, KP., Shi, Y., Jing, X. et al. (2016). Effects of Short-Term Warming and Altered Precipitation on Soil Microbial Communities in Alpine Grassland of the Tibetan Plateau. Frontiers in Microbiology, 7, 1-11.

Ma ZY., Liu, HY., Mi, ZR. et al. (2017). Climate warming reduces the temporal stabilityof plant community biomass production. Nature Communications, 8, 15378.

Liu, HY., Mi, ZR., Lin, L. et al. (2018). Shifting plant species composition in response to climate change stabilizes grassland primary production. Proceedings of the National Academy of Sciences, 115, 4051-4056.

The qSIP calculations reported in the methodology for this work are rather superficial and the reader must be experienced in this technique to understand how the incorporators were identified and their growth quantified. For instance, the GC content of taxa was calculated for reads clustered in OTUs, and it is not discussed in the text the validity of such approach working at genus level.

We have added the description of qSIP calculations in Supplementary Materials.

The approach of GC content calculation can be used at genus level (Koch et al., 2018). The GC content of each bacterial taxon (Gi) was calculated using the mean density for the unlabeled (WLIGHTi) treatments (Hungate et al. 2015), rather than OTU sequence information. We have revised the sentence in MATERIALS AND METHODS, that is “the number of 16S rRNA gene copies per OTU taxon (e.g., genus or OTU) in each density fraction was calculated by multiplying the relative abundance (acquisition by sequencing) by the total number of 16S rRNA gene copies (acquisition by qPCR)” (Line 255-258).

Reference:

Hungate, B., Mau, R., Schwartz, E., Caporaso, J., Dijkstra, P., Van Gestel, N. et al. (2015). Quantitative microbial ecology through stable isotope probing. Applied and Environmental Microbiology, 81, 7570-7581.

Koch, B., McHugh, T., Hayer, M., Schwartz, E., Blazewicz, S., Dijkstra, P. et al. (2018). Estimating taxon-specific population dynamics in diverse microbial communities. Ecosphere, 9, e02090.

The selection of V4-V5 region over V3-V4 region to quantify the number of copies of the 16S rRNA gene should be substantiated in the text. Classic works determined one decade ago that primer pairs that amplify V3-V4 are most suitable to assess soil bacterial communities. Hungate et al. (2015), worked with the V3-V4 region when establishing the qSIP method. Maybe the number of unassigned OTUs is related with the selection of this region.

Both primer sets (V3-V4 and V4-V5 regions), are widely used across various sample sets, with highly similar in representing the total microbial community composition (Fadeev et al., 2021; Zhang et al., 2018).

A previous study based on our Field Research Station of Alpine Grassland Ecosystem used V4-V5 primer pairs to investigated the effect of warming and altered precipitation on the overall bacterial community composition (Zhang et al., 2016).

Another reason for choosing the V4-V5 primer set in this study was to integrate and compare the data with that of two previous qSIP studies (Ruan et al., 2023; Guo et al., submitted), both of them focused on the growth responses of active species to global change and used V4-V5 primer pairs.

We have added an explanation about primer selection as “The V4-V5 primer pairs were chosen to facilitate integration and comparison with data from previous studies (Ruan et al., 2023; Zhang et al., 2016)” (Line 213-215).

Reference:

Fadeev, E., Cardozo-Mino, M.G., Rapp, J.Z. et al. (2021). Comparison of Two 16S rRNA Primers (V3–V4 and V4–V5) for Studies of Arctic Microbial Communities. Frontiers in Microbiology, 12

Zhang, J.Y., Ding, X., Guan, R. et al. (2018). Evaluation of different 16S rRNA gene V regions for exploring bacterial diversity in a eutrophic freshwater lake. Science of The Total Environment, 618, 1254-1267.

Zhang, K.P., Shi, Y., Jing, X. et al. (2016). Effects of Short-Term Warming and Altered Precipitation on Soil Microbial Communities in Alpine Grassland of the Tibetan Plateau. Frontiers in Microbiology, 7, 1-11.

Ruan, Y., Kuzyakov, Y., Liu, X. et al. (2023). Elevated temperature and CO2 strongly affect the growth strategies of soil bacteria. Nature Communications, 14, 1-12.

Guo, J.J., Kuzyakov, Y., Li, L. et al. (2023). Bacterial growth acclimation to long-term nitrogen input in soil. The ISME Journal, Submitted.

Report of preprocessing and processing of the sequences does not comply state of the art standards. More info on how the sequences were handled is needed, taking into account that a significant part of the manuscript relies on taxonomic classification of such sequences. Also, an OTU approach for an almost species-dependent analysis (GC contents) should be replaced or complemented with an ASV or subOTUs approach, using denoisers such as DADA2 or deblur. Usage of functional prediction tools underestimates gene frequencies, including those related with biogeochemical significance for soil-carbon and nitrogen cycling.

(1) We have complemented the information about sequence processing as “The raw sequences were quality-filtered using the USEARCH v.11.0 (Edgar, 2010). In brief, the paired-end sequences were merged and quality filtered with “fastq_mergepairs” and “fastq_filter” commands, respectively. Sequences < 370 bp and total expected errors > 0.5 were removed. Next, “fastx_uniques” command was implemented to remove redundant sequences. Subsequently, high-quality sequences were clustered into operational taxonomic units (OTUs) with “cluster_otus” commandat a 97% identity threshold, and the most abundant sequence from each OTU was selected as a representative sequence.” (Line 238-245).

(2) We have complemented the zero-radius OTU (ZOTU) analysis by the unoise3 command in USEARCH (https://drive5.com/usearch/manual/pipe_otus.html), as shown in Fig. S1-S2. The results showed that overall growth responses of soil bacteria to warming and precipitation changes were similar based on OTU and ZOTU analyses, i.e., warming and altered precipitation tend to negatively affect the growth of grassland bacteria and the prevalence of antagonistic interactions of T and P. The similarity of results between the different methods is reflected at the overall community level, the phylum level, the genus level and the species (i.e., OTU or ZOTU) level (Fig. S1 and S2).

**Author response image 1. sa2fig1:** The growth responses of grassland bacteria to warming and altered precipitation based on ZOTU analysis. The results of growth rates at the community level (**A**), the phylum level (**B**), and the ZOTU level (**C and D**) were similar to those based on OTU analysis. (**C**) The single and combined factor effects of climate factors on species growth, by comparing with the growth rates in T0nP. (**D**) The proportions of species growth influenced by different interaction types of T and P. T^0^-P represents the ambient temperature and decreased precipitation; T^+^-P represents warming and decreased precipitation; T^0^cP represents ambient temperature and precipitation; T^+^cP represents warming and ambient precipitation; T^0^+P represents ambient temperature and enhanced precipitation; T^+^+P represents warming and enhanced precipitation. Values represent mean and the error bars represent standard deviation. Different letters indicate significant differences between climate treatments.

**Author response image 2. sa2fig2:** The growth responses of grassland bacteria at the genus level to warming and altered precipitation based on OTU analysis (**A and C**) and ZOTU analysis (**B and D**). (**A and B**) The single and combined factor effects of climate factors on growth in genera, by comparing with those in T0nP. (**C and D**) The proportions of genera whose growth influenced by different interaction types of T and P.

(3) Agreed. We have introduced the caveat about the limitation of usage of functional prediction tools to the end of DISCUSSION, that is “This is, however, still to be verified, as the functional output from PICRUSt2 is less likely to resolve rare environment-specific functions (Douglas et al., 2020)” (Line 540-542). The caveat ensures that the reader knows the limitations of these methods, and are not potentially overstate our findings.

Reference:

Douglas, G.M., Maffei, V.J., Zaneveld, J.R. et al. (2020) PICRUSt2 for prediction of metagenome functions. Nat Biotechnol. 38, 685–688.

**Reviewer #2 (Recommendations For The Authors):**
General suggestions:Regarding the qSIP method, would be of help to see the differences in density vs number of 16S rRNA gene abundance for the most responsive bacterial groups in the different treatments, taking into account that with only 7 fractions the entire change in bacterial growth was resolved.

We have selected three representative bacterial taxa (OTU1 belonging to Bradyrhizobium, OTU14 belonging to Solirubrobacter, OTU15 belonging to Pseudoxanthomonas), which have high growth rates in climate change treatments. The result showed that the peaks in the 18O treatment are shifted "backwards" (greater average weighted buoyancy density) compared to the 16O treatment, indicating that these species assimilates the 18O isotope into the DNA molecules during growth.

**Author response image 3. sa2fig3:** The distribution of 16S rRNA gene abundance of three representative bacterial taxa (OTU1- Bradyrhizobium, OTU14-Solirubrobacter, and OTU15-Pseudoxanthomonas) in different buoyant density fractions. Values represent mean and the error bars represent standard deviation.

Seven fractionated DNA samples were selected for sequencing because they contained more than 99% gene copy numbers of each samples (please see the Figure below). The DNA concentrations of other fractions were too low to construct sequencing libraries.

**Author response image 4. sa2fig4:** Relative abundance of 16S rRNA gene copies in each fraction. The fractions with density between 1.703 and 1.727 g ml-1 were selected because they contained more than 99% gene copy numbers of each sample. T0-P represents the ambient temperature and decreased precipitation; T+-P represents warming and decreased precipitation; T0cP represents ambient temperature and precipitation; T+cP represents warming and ambient precipitation; T0+P represents ambient temperature and enhanced precipitation; T++P represents warming and enhanced precipitation. Values represent mean and the error bars represent standard deviation.

With such dataset additional multivariate analysis would be of help to better interpret the ecological framework.

Thanks for the suggestion. Interpreting the ecological framework is meaningful for understanding microbial responses to environmental changes.

The main objective of this study is to investigate the growth response of soil microbes in alpine grasslands to the temperature and precipitation changes, and the interaction between climate factors. Our results, as well as the results of complementary analyses (based on subOTU analyses, SHOWN BELOW), indicate that warming and altered precipitation tend to negatively affect the growth of grassland bacteria, and the prevalence of antagonistic interactions of T and P.

We have emphasized our research objectives and main conclusions in the revised manuscript: “The goal of current study is to comprehensively estimate taxon-specific growth responses of soil bacteria following a decade of warming and altered precipitation manipulation on the alpine grassland of the Tibetan Plateau” (Line 112-114);

“Our results demonstrated that both warming and altered precipitation negatively affect the growth of grassland bacteria; However, the combined effects of warming and altered precipitation on the growth of ~70% soil bacterial taxa were smaller than the single-factor effects, suggesting antagonistic interaction” (Line 552-556).

Extension of interaction analysis and its conclusions should be shortened, summarizing the most relevant findings. In my opinion, it becomes a bit redundant.

We have shortened the discussion of Extension of interaction analysis by deleting the little relevant contents.

Below are some, but not all, examples that have been deleted or revised in the Discussion,

(1) Deleted “This result supports our second hypothesis that the interactive effects between warming and altered precipitation on soil microbial growth are not simply additive”;

(2) Deleted “A previous study suggested that multiple global change factors had negative effects on soil microbial diversity (Yang et al., 2021)”;

(3) Revised “A meta‐analysis of experimental manipulation revealed that the combined effects of different climate factors were usually less than expected additive effects, revealing antagonistic interactions on soil C storage and nutrient cycling processes (Dieleman et al., 2012; Wu et al., 2011). Moreover, two experimental studies on N cycling and net primary productivity demonstrated that the majority of interactions among multiple factors are antagonistic rather than additive or synergistic, thereby dampening the net effects (Larsen et al., 2011; Shaw et al., 2002)” to “A range of ecosystem processes have been revealed to be potentially subject to antagonistic interactions between climate factors, for instance, net primary productivity (Shaw et al., 2002), soil C storage and nutrient cycling processes (Dieleman et al., 2012; Wu et al., 2011; Larsen et al., 2011)” (Line 499-503);

(4) Revised “Previous evidences suggest that warming has a negative impact on soil carbon pools (Jansson & Hofmockel, 2020; Purcell et al., 2022). During the first phase of soil warming (~ 10 years), microbial activity increased, resulting in rapid soil carbon mineralization and respiration (Melillo et al., 2017)” to “Previous evidences suggest that warming has a negative impact on soil carbon pools (Jansson & Hofmockel, 2020; Purcell et al., 2022), mainly because of the rapid soil carbon mineralization and respiration (Melillo et al., 2017)” (Line 464-466).

I strongly suggest a functional analysis based on shotgun sequencing or RNAseq approaches. With this approach this work would be able to answer who is growing under altered T and Precipitation regimes and what are those that are growing doing.

Thanks for the suggestion. Metagenomic sequencing is a popular approach to evaluate potential functions of microbial communities in environment. However, there are two main reasons that limit the application of metagenomic or metatranscriptomic sequencing in this study: (1) Most of the fractionated samples in SIP experiment have low DNA concentration and do not meet the requirement of library construction for sequencing; (2) Metagenome and metatranscriptomics usually have relatively low sensitivity to rare species, reducing the diversity of detected active species.

This study focused on active microbial taxa and their growth in response to multifactorial climate change. We have added the prospect in DISCUSSION, that is “This suggests the development of methods combining qSIP with metagenomes and metatranscriptomes to assess the functional shifts of active microorganisms under global change scenarios” (Line 542-544).

Minor suggestions:L121. _As

We have deleted this sentence and relocated the hypotheses in the last paragraph of INTRODUCTION (according to the suggestion of the reviewer #3).

Line150. Described previously in.

Done (Line 136).

Line500. Check whether it is better to use the word acclimatization (Coordinated response to several simultaneous stressors) in exchange of acclimation

We have revised it according to this suggestion (Line 481).

Fig.4C Drought

Done (Line 761).

**Reviewer #3 (Public Review):**
Summary:In this paper, Ruan et al. studied the long-term impact of warming and altered precipitations on the composition and growth of the soil microbial community. The researchers adopted an experimental approach to assess the impact of climate change on microbial diversity and functionality. This study was carried out within a controlled environment, wherein two primary factors were assessed: temperature (in two distinct levels) and humidity (across three different levels). These factors were manipulated in a full factorial design, resulting in a total of six treatments. This experimental setup was maintained for ten years. To analyze the active microbial community, the researchers employed a technique involving the incorporation of radiolabeled water into biomolecules (particularly DNA) through quantitative stable isotope probing. This allowed for the tracking of the active fraction of microbes, accomplished via isopycnic centrifugation, followed by Illumina sequencing of the denser fraction. This study was followed by a series of statistical analysis to identify the impact of these two variables on the whole community and specific taxonomic groups. The full factorial design arrangement enabled the researchers to discern both individual contributions as well as potential interactions among the variablesStrengths:This work presents a timely study that assesses in a controlled fashion the potential impact of global warming and altered precipitations on microbial populations. The experimental setup, experimental approach and data analysis seem to be overall solid. I consider the paper of high interest for the whole community as it provides a baseline to the assessment of global warming on microbial diversity.

Thanks for the encouragement and positive comments.

Weaknesses:While taxonomic information is interesting, it would have been highly valuable to include transcriptomics data as well. This would allow us to understand what active pathways become enriched under warming and altered precipitations. Non-metabolic OTUs hold significance as well. The authors could have potentially described these non-incorporators and derived hypotheses from the gathered information. The work would have benefited from using more biological replicates of each treatment.

Thanks for the valuable suggestions.

(1) Metatranscriptomics can assess the functional profiles of the community, but it has relatively low sensitivity to rare species, which is difficult to correlate the function pathways with the assignment to the numerous active taxa identified by qSIP. Additionally, due to the low DNA concentration, most fractionated samples are difficult to construct sequencing libraries, while amplicon based sequencing analyses were allowed. This study therefore focused on active microbial taxa and their growth in response to multifactorial climate change. We have added the prospect in DISCUSSION, that is “This suggests the development of methods combining qSIP with metagenomes and metatranscriptomes to assess the functional shifts of active microorganisms under global change scenarios” (Line 542-544).

(2) 18O-qSIP can identify the growing microbial species (i.e., 18O incorporators) in the environment rather than metabolically active taxa. These non-incorporators in our study were likely to be metabolically active, i.e., maintaining life activities without reproduction, or recently deceased (Blazewicz et al., 2013). Therefore, it is hard to distinguish whether these non-incorporators possess metabolic activity.

(3) Agreed. The qSIP experiments involve the use of isotopes and the sequencing of a large number of DNA samples (90 samples per biological replicate in this study). Considering its high cost, we selected three replicates for analysis. We have explained this issue in MATERIALS AND METHODS, that is “Considering the cost of qSIP experiment (i.e., the use of isotopes and the sequencing of a large number of DNA samples), we randomly selected three out of the six plots, serving as three replicates for each treatment” (Line 154-157).

Reference:

Nuccio, E.E., Starr, E., Karaoz, U. et al. (2020) Niche differentiation is spatially and temporally regulated in the rhizosphere. ISME J 14, 999–1014.

Blazewicz, S.J., Barnard, R.L., Daly, R.A., Firestone, M.K (2013). Evaluating rRNA as an indicator of microbial activity in environmental communities: limitations and uses. The ISME Journal, 7, 2061–2068.

**Reviewer #3 (Recommendations For The Authors):**
Major comments:The manuscript should be written in a clearer way. The language should be more direct, so the message is conveyed faster and clearer. Some sentences, for instance, could be shortened or re-organized. Below, you will find some examples.

We have rewritten the sentences to make the manuscript clearer. Below are some, but not all, examples that have been revised:

(1) Deleted “(reduced precipitation, hereafter ‘drought’, or enhanced precipitation, hereafter ‘wet’)” in INTRODUCTION;

(2) Deleted “Controlled experiments simulating climate change have investigated changes in microbial community composition as measured by shifts in the relative abundances (Evans & Wallenstein, 2014; Barnard et al., 2015). However, changes in relative abundances may be poor indicators of population responses to environmental change in some cases (Blazewicz et al., 2020). Another challenge is the presence of a large number of inactive microbial cells in the soil, which hinders the direct, quantitative measure of the ecological drivers in population dynamics (Fierer, 2017; Lennon & Jones, 2011).” in DISCUSSION;

(3) Deleted “This result supports our second hypothesis that the interactive effects between warming and altered precipitation on soil microbial growth are not simply additive” in DISCUSSION;

(4) Deleted “A previous study suggested that multiple global change factors had negative effects on soil microbial diversity (Yang et al., 2021)” in DISCUSSION;

(5) Revised “A meta‐analysis of experimental manipulation revealed that the combined effects of different climate factors were usually less than expected additive effects, revealing antagonistic interactions on soil C storage and nutrient cycling processes (Dieleman et al., 2012; Wu et al., 2011). Moreover, two experimental studies on N cycling and net primary productivity demonstrated that the majority of interactions among multiple factors are antagonistic rather than additive or synergistic, thereby dampening the net effects (Larsen et al., 2011; Shaw et al., 2002)” to “A range of ecosystem processes have been revealed to be potentially subject to antagonistic interactions between climate factors, for instance, net primary productivity (Shaw et al., 2002), soil C storage and nutrient cycling processes (Dieleman et al., 2012; Wu et al., 2011; Larsen et al., 2011)” in DISCUSSION (Line 499-503);

(6) Revised “Previous evidences suggest that warming has a negative impact on soil carbon pools (Jansson & Hofmockel, 2020; Purcell et al., 2022). During the first phase of soil warming (~ 10 years), microbial activity increased, resulting in rapid soil carbon mineralization and respiration (Melillo et al., 2017)” to “Previous evidences suggest that warming has a negative impact on soil carbon pools (Jansson & Hofmockel, 2020; Purcell et al., 2022), mainly because of the rapid soil carbon mineralization and respiration (Melillo et al., 2017)” in DISCUSSION (Line 464-466).

I'm curious about why, even though there were six replicates of the experiment, only three samples were collected for analysis. Metagenomic analyses tend to display high variability.

The qSIP experiments involve the use of isotopes and the sequencing of a large number of DNA samples (90 samples per biological replicate in this study). Considering its high cost, we selected three replicates for analysis..

In Fig. 3A, the absolute growth rates (16S copies/d*g) are shown. How do you know that the efficiency of DNA extraction was similar across all treatments and therefore the absolute numbers are comparable?

To avoid differences in extraction efficiency caused by experimental procedures, all DNA samples were extracted by the same person (the first author) within 2-3 hours, and a unifying procedure of cell lysis and DNA extraction was used, i.e., the mechanical cell destruction was attained by multi-size beads beating at 6 m s-1 for 40 s, and then FastDNA SPIN Kit for Soil (MP Biomedicals, Cleveland, OH, USA) was used for DNA extraction.

We have measured the concentration of extracted DNA and found no significant difference between treatments (Table for the response letter).

**Author response table 1. sa2table1:** Soil DNA concentration in climate change treatments after qSIP incubation (measured by Qubit DNA HS Assay Kits).

ng//mul	T^(0)-P	T^(+)-P	T^(0)cP	T^(+)cP	T^(0)+P	T^(+)+P	F value	p value
^(16) O-treatment	98+-11	100+-11	98+-19	109+-11	95+-17	93+-20	0.34	0.88
^(18) O-treatment	105+-21	99+-13	83+-24	98+-5	75+-13	78+-18	1.49	0.27
F value	0.21	0.02	0.71	2.17	2.47	0.92		
p value	0.7	0.9	0.45	0.22	0.19	0.39		

Values represent mean and standard deviation. T0-P represents the ambient temperature and decreased precipitation; T+-P represents warming and decreased precipitation; T0cP represents ambient temperature and precipitation; T+cP represents warming and ambient precipitation; T0+P represents ambient temperature and enhanced precipitation; T++P represents warming and enhanced precipitation. The results of ANOVA indicated no significant difference of extracted DNA concentration between treatments (p > 0.05).

We have introduced the caveat in the DISCUSSION, that is “Note that the experimental parameters such as DNA extraction and PCR amplification efficiencies also have significant effects on the accuracy of growth assessment. This alerts the need to standardize experimental practices to ensure more realistic and reliable results” (Line 544-547).

Line 96-99 and 121-124: "Hypotheses are typically placed at the end of the final paragraph in the Introduction section. It is advisable to relocate them there and provide a clearer description of the paper's main goal."

We have relocated the hypotheses at the end of INTRODUCTION, and the main goal of this study, that is “The goal of current study is to comprehensively estimate taxon-specific growth responses of soil bacteria following a decade of warming and altered precipitation manipulation on the alpine grassland of the Tibetan Plateau, by using the 18O-quantitative stable isotope probing (18O-qSIP)” (Line 112-115).

Line 399: Although you describe the classification among antagonistic interactions in the Methods section, I think you should describe this in further detail here. Can you clarify how you carried out this categorization and how these results were interpreted considering the phylogenetic classification.

We have added the description of antagonistic interactions, that is “The interaction type of T and P on the growth of ~70% incorporators was antagonistic (i.e., the combined effect size is smaller than the additive expectation) (Fig. 4C)” (Line 388-390).

The interaction types between factors can be classified into three categories: additive, synergistic and antagonistic. Additive interactions are those in which the combined effect size of factors is equal to the sum of the single effects of the factors (i.e., additive expectation, Fig. 1B). Synergistic interactions refer to the effect size was larger than the additive expectation by the combined manipulation of factors. On the contrary, antagonistic interactions refer to the combined effect size of factors is smaller than the additive expectation. In this study, the antagonistic interactions were further divided into three sub-categories: weak antagonistic interaction, strong antagonistic interaction, and neutralizing effect (Fig. 1B). The weak antagonistic interaction refers to the combined effect size smaller than the additive expectation and larger than any of the single factor effects. The strong antagonistic interaction refers to that the combined effect size is smaller than any of the single factor effects but larger than 0. The neutralizing effect refers to that the combined effect size is equal to 0, implying that the effects of different factors cancel each other out.

Methodologically, the single and combined effects of two climate factors and their interaction effects were calculated by the natural logarithm of response ratio (lnRR) and Hedges’ *d*, respectively (Yue et al., 2017).

We have added the result interpretation about the phylogenetic distribution patterns of incorporators, that is “The degree of phylogenetic relatedness can indicate the processes that influenced community assembly, like the extent a community is shaped by environmental filtering (clustered by phylogeny) or competitive interactions (life strategy is phylogenetically random distribution) (Evans & Wallenstein, 2014; Webb et al., 2002).The results showed that the incorporators whose growth was influenced by the antagonistic interaction of T and P showed significant phylogenetic relatedness, indicating the occurrence of taxa more likely shaped by environment filtering (i.e., selection pressure caused by changes in temperature and moisture conditions). In contrast, the growing taxa affected by synergistic interactions of T and P showed random phylogenetic distributions (Table S1), which may be explained by competition between taxa with similar eco-physiological traits or changes in genotypes (possibly through horizontal gene transfer) (Evans & Wallenstein, 2014). We also found that the extent of phylogenetic relatedness to which taxa groups of T and P interaction types varied by climate scenarios, suggesting that different climate history processes influenced the ways bacteria survive temperature and moisture stress” (Line 515-529).

Reference:

Evans, S.E. & Wallenstein, M.D. (2014). Climate change alters ecological strategies of soil bacteria. Ecology Letters, 17, 155-164.

Webb, C.O., Ackerly, D.D., McPeek, M.A. & Donoghue, M.J. (2002). Phylogenies and Community Ecology. Annual Review of Ecology and Systematics, 33, 475-505.

Yue, K., Fornara, D.A., Yang, W., Peng, Y., Peng, C., Liu, Z. et al. (2017). Influence of multiple global change drivers on terrestrial carbon storage: additive effects are common. Ecology Letters, 20, 663-672.

Line 407-8: What do you mean with "...clustered at the phylogenetic branches" and Line 410: "cluster near the tips of the phylogenetic tree". Can you please clarify?

Sorry for the unclear statement. We have added the explanation of NTI, that is “Nearest taxon index (NTI) was used to determine whether the species in a particular growth response are more phylogenetically related to one another than to other species (i.e., close or clustering on phylogenetic tree). NTI is an indicator of the extent of terminal clustering, or clustering near the tips of the tree (Evans & Wallenstein, 2014; Webb et al., 2002)” (Line 397-401).

Reference:

Evans, S.E. & Wallenstein, M.D. (2014). Climate change alters ecological strategies of soil bacteria. Ecology Letters, 17, 155-164.

Webb, C.O., Ackerly, D.D., McPeek, M.A. & Donoghue, M.J. (2002). Phylogenies and Community Ecology. Annual Review of Ecology and Systematics, 33, 475-505.

Could you provide some info about the biochemistry of the incorporation of heavy water into DNA molecules? What specific enzymes are typically involved?

Due to the low DNA concentration in most fractionated samples (less than 10 ng/μL, measured by Qubit DNA HS Assay Kits), only amplicon based sequencing analyses were allowed. This study therefore focused only on active microbial taxa and their growth in response to multifactorial climate change.

What might be the impact of soil desiccation on bacterial survival and subsequent water uptake?

Slow dehydration and air drying of soil is a very common phenomenon in nature (Koch et al., 2018). In this process, microorganisms will reduce metabolism, and shift towards a potentially active state (Blagodatskaya and Kuzyakov, 2013). A previous study suggested that the potentially active microbial population permanently existing in soil between the active and dormant physiological states. Even under long-term starvation the potentially active microorganisms maintain ‘physiological alertness’ to be ready to occasional substrate input (Blagodatskaya and Kuzyakov, 2013). These microorganisms are important participants in the biogeochemical cycle is the focus of this study.

Replacing the environmental water in the soil with 18O-labelled water is a typical practice for qSIP studies (Hungate et al. 2015; Koch et al., 2018). This process may cause disturbance to the microbial community. In this study, the soil samples were placed in a thermostatic incubator (14℃ and 16℃), rather than air-drying at 25℃ (as used in most studies). The incubation temperature is relatively low (compared to 25℃) and there is no violent air convection in the incubator, resulting slower evaporation and no significant discoloration caused by severe soil dehydration after 48 h. The process of soil drying in this study simulated the natural phenomenon, i.e., slow water loss in soil.

We have added the description in MATERIALS AND METHODS, that is “There is no violent air convection in the incubator and the incubation temperature is relatively low (compared to 25℃ used in previous studies), resulting slower evaporation and no significant discoloration caused by severe soil dehydration after 48 h” (Line 171-174).

Reference:

Blagodatskaya, E. & Kuzyakov, Y. (2013) Active microorganisms in soil: Critical review of estimation criteria and approaches. Soil Biology and Biochemistry, 67, 192-211.

Hungate, B., Mau, R., Schwartz, E., Caporaso, J., Dijkstra, P., Van Gestel, N. et al. (2015). Quantitative microbial ecology through stable isotope probing. Applied and Environmental Microbiology, 81, 7570-7581.

Koch, B., McHugh, T., Hayer, M., Schwartz, E., Blazewicz, S., Dijkstra, P. et al. (2018). Estimating taxon-specific population dynamics in diverse microbial communities. Ecosphere, 9, e02090.

The analysis of the 180 incorporators is interesting as it defines what microbes are metabolically active and hence growing under the different conditions tested. Should not be worth to analyze the non-incorporators? Is it possible to identify a pattern to generate a hypothesis of why they are metabolically inactive based on this information? In the Methods section, the authors state that they identified a total of 6,938 OTUs, of which only 1,373 were found to be incorporators.

Microbes exist in a range of metabolic states: growing, active (non-growth), dormant and recently deceased (Blazewicz et al., 2013), and there is still a lack of clear threshold for their identification. 18O-DNA qSIP can identified the growing microbial species (i.e., 18O incorporators) rather than all metabolic active taxa, because some cells are measurably metabolizing (catabolic and/or anabolic processes) without reproduction. Therefore, the non-incorporators in our study may be metabolically active, or not (recently deceased microorganisms). This study focuses on the growing microorganisms identified by 18O-qSIP.

In this study, ~20% microbial taxa (1,373/6,938) were identified as 18O incorporators. Microorganisms in soils suffer from resource and energy constraints frequently (Blagodatskaya and Kuzyakov, 2013). The energy requirements of species in the growing state are much higher (~30 fold) than those in the non-growing state, so the percentage of growing bacterial taxa in soil tends to be low.

Reference:

Blazewicz, S.J., Barnard, R.L., Daly, R.A., Firestone, M.K (2013). Evaluating rRNA as an indicator of microbial activity in environmental communities: limitations and uses. The ISME Journal, 7, 2061–2068.

Blagodatskaya, E. & Kuzyakov, Y. (2013) Active microorganisms in soil: Critical review of estimation criteria and approaches. Soil Biology and Biochemistry, 67, 192-211.

Minor comments:Fig. 3A and 3B. Please show the results of the multiple comparisons.

Done.

**Author response image 5. sa2fig5:** Bacterial growth responses to climate change and the interaction types between warming and altered precipitation. The growth rates (**A**), and responses (LnRR) of soil bacteria to warming and altered precipitation (**B**) at the whole community level. The growth rates (**C**), and responses of the dominant bacterial phyla (**D**) had similar trends with that of the whole community. Values represent mean and the error bars represent standard deviation. Different letters indicate significant differences between climate treatments.

Fig. 4. This figure should be self-explanatory. This diagram is challenging to understand.

We have revised Fig. 4 to improve clarity.

**Author response image 6. sa2fig6:** The growth responses and phylogenetic relationship of incorporators subjected to different interaction types under two climate scenarios. A phylogenetic tree of all incorporators observed in the grassland soils (**A**). The inner heatmap represents the single and combined factor effects of climate factors on species growth, by comparing with the growth rates in T0nP. The outer heatmap represents the interaction types between warming and altered precipitation under two climate change scenarios. The proportions of positive or negative responses in species growth to single and combined manipulation of climate factors by summarizing the data from the inner heatmap (**B**). The proportions of species growth influenced by different interaction types of T and P by summarizing the data from the outer heatmap (**C**).

Fig. 4. It says "Dorought" instead of "drought"

Done (Line 760).

Line 109: "relieves" instead of "relieved"

Done (Line 102).

Line 129: Should be: "We classified the interaction types as additive, synergistic, antagonistic, null and neutralizing."

Done (Line 117).

Line 233: How were the 16S rRNA sequences from each density fraction analyzed?

(1) Raw sequencing data processing:

The raw 16S rRNA gene sequences of each density fraction were quality-filtered using the USEARCH v.11.0 (Edgar, 2010). The paired-end sequences were merged and quality filtered with “fastq_mergepairs” and “fastq_filter” commands, respectively. Sequences < 370 bp and total expected errors > 0.5 were removed. Next, “fastx_uniques” command was implemented to identify the unique sequences. Subsequently, high-quality sequences were clustered into operational taxonomic units (OTUs) with “cluster_otus” commandat a 97% identity threshold, and the most abundant sequence from each OTU was selected as a representative sequence. The taxonomic affiliation of the representative sequence was determined using the RDP classifier (Wang et al., 2007).

(2) qSIP calculation:

Sequencing data reflects the relative abundance of taxa in community. We multiply the OTU’s relative abundance (acquisition by sequencing) and the number of 16S rRNA gene copies (acquisition by qPCR) to obtain the number of gene copies per OTU in each fraction. Then, the proportion of gene copies of a specific OTU of each fraction relative to the total amount of gene copies in one sample was calculated and used as a weight value for further calculation of the average weighted buoyant density (the critical parameter for assessing microbial growth).

Line 366: "Three single-factor ... between warming and altered precipitation" -> "The individual impact of warming, drought, and wet conditions resulted in the most substantial negative effects on bacterial growth compared with the effects of warming x drought and warming x wet. A result that illustrates the negative interactions between warming and modified precipitations patterns."

Done (Line 365-368).

Line 376: "Similar with the result of whole growth of bacteria community, the growth responses of the major bacterial phyla were also negatively influenced by single climate factors". This sentence is hard to read. Maybe something like this: "Growth of the major bacterial phyla was also negatively influenced by the individual climate factors".

Done (Line 371-372).

Line 383: "In particular, the effects of wet and warming neutralized each other, resulting the net effects became zero on the growth rates of the phyla Actinobacteria and Bacteroidetes". "In Actinobacteria and Bacteroidetes, the effect of wet and warming neutralized each other, as the combined effect of these two factors had no effect on growth".

Done (Line 377-379).

Line 390: "The individual warming treatment (T+nP) reduced the growth rates of 75% incorporators..." "Warming (T+nP) reduced the growth of 75% of the taxonomic groups, which was followed by drought and wet.

Done (Line 384-385).

Line 392: "The combined manipulations of warming and altered precipitation lowered the percentages of incorporators with negative responses compared with single factor manipulation, especially warming and enhanced precipitation manipulation" -> "Warming x drought and warming x wet had a smaller impact on the growth of incorporators, compared with single effects."

Done (Line 385-387).

Line 468. This sentence "To the best ..." is not necessary.

We have deleted this sentence.

Line 476. Is it really "synthesis" the word you want to use?

We have deleted this sentence.

Line 477. Maybe should written like this: "Consistent with our findings, a recent experimental study demonstrated that 15 years of warming reduced the growth rate of soil bacteria in a montane meadow in northern Arizona."

Done (Line 459-461).

Line 490 and 502. Consider using "however" only once in a paragraph.

We have deleted the second “however” (Line 483).

Line 555-559. Based on genomic data you cannot predict the functional role of microbes in the environment. These sentences are speculative. Please, consider using less strong affirmations and focus more on the pathways that are enriched in the incorporators.

Agreed. We have deleted this part of content.